# Towards Autonomous Experimentation: BioProBench, a Corpus and Benchmark for Biological Protocol Comprehension

## Abstract

The automation of scientific experimentation is critically hindered by the inability of Large Language Models (LLMs) to reliably comprehend the specialized, accuracy-critical, and procedural nature of biological protocols. To address this fundamental challenge, we present **BioProBench**, a comprehensive resource for procedural reasoning in biology. BioProBench is grounded in a foundational Bio-ProCorpus of 27,000 human-written protocols. From this corpus, we systematically constructed a dataset of over 550,000 task instances, partitioning it into a large-scale training set and a rigorous benchmark with a held-out test set and novel evaluation metrics. Our comprehensive evaluation of 10 mainstream LLMs on the benchmark reveals a critical performance gap: while models excel on basic comprehension tasks, they underperform on tasks requiring deep procedural logic, quantitative accuracy, and safety-critical reasoning. To demonstrate the value of BioProCorpus in mitigating these issues, we developed **ProAgent**, a Retrieval-Augmented Generation (RAG) agent. Grounded in our corpus, ProAgent substantially advances the state-of-the-art. BioProBench thus provides both a rigorous diagnostic benchmark and a foundational resource for developing the next generation of reliable AI for science. The code and data are available at: `https://anonymous.4open.science/r/Anonymization-112358`.

## 1 Introduction

Biological experimental protocols, comprehensive documents detailing reagents, instruments, and crucially, step-by-step procedures, form the backbone of life science research. With laboratories increasingly adopting high-throughput automation (Murthy & Lim, 2024) and cloud-based execution platforms (Boiko et al., 2023), the demand for reliable, automated interpretation of these protocols has become critical (Ren et al., 2025). However, the procedural, causal, and safety-critical nature of this domain presents a formidable challenge for Large Language Models (LLMs). Minor misinterpretations can lead to experimental failure, resource wastage, or unsafe laboratory conditions.

While LLMs have shown significant progress in general biomedical text mining, existing models and benchmarks (Nori et al., 2023; Thirunavukarasu et al., 2023; Liévin et al., 2024; Yang et al., 2024; Singhal et al., 2025) primarily focus on declarative knowledge (*e.g.*, summarizing research findings from articles). Specialized models like BioBERT (Lee et al., 2020), BioGPT (Luo et al., 2022), and BioMedGPT (Luo et al., 2023) demonstrate domain-specific adaptation by fine-tuning on biomedical corpora. Existing biomedical text processing benchmarks, including BioASQ (Tsatsaronis et al., 2015), PubMedQA (Jin et al., 2019), LAB-Bench (Laurent et al., 2024) and BixBench (Mitchener et al., 2025) primarily focus on question answering and data interpretation. They often fall short in comprehending procedural knowledge, the structured, causal, and conditional logic that defines an experiment. This gap is a primary bottleneck for true scientific automation.

To fill this critical gap, we first introduce the **BioProBench**, a comprehensive resource for evaluating and improving procedural reasoning in biological contexts (Figure 1). BioProBench contains: (1) a foundational **corpus** of 27,000 professionally authored protocols; (2) a structured **dataset** of over 550,000 instances derived from this BioProCorpus, which is partitioned into a training set to facilitate model fine-tuning and a held-out test set; and (3) a rigorous **benchmark** with a novel, domain-

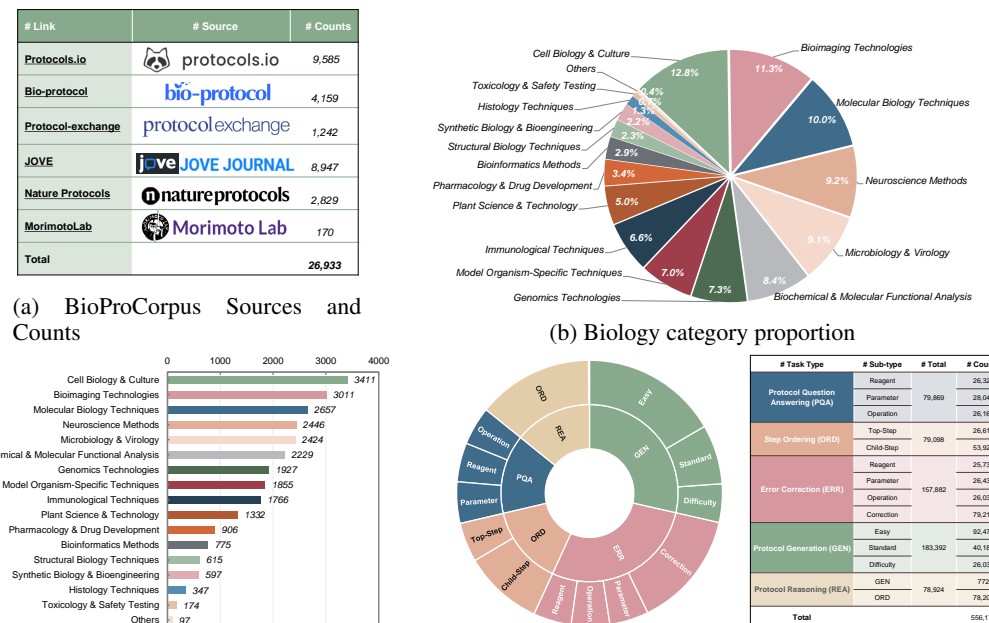

(a) BioProCorpus Sources and Counts

(b) Biology category proportion

(c) Biology Category Counts

(d) Task and Sub-Task Counts

Figure 1: Overview of the BioProBench.

specific metrics to evaluate procedural understanding, such as *keyword-based content metrics* and *embedding-based structural metrics*, to accurately quantify procedural knowledge. Our evaluation of 10 state-of-the-art LLMs on the BioProBench benchmark reveals that the top-tier models excel at basic comprehension, yet their effectiveness degrades significantly on tasks requiring deep procedural logic. We further demonstrate the BioProCorpus's practical utility by developing **ProAgent**, a retrieval-augmented agent that substantially improves reasoning accuracy and procedural step recall, charting a path toward more reliable AI for science.

Overall, our contributions are summarized as follows:

- We present **BioProBench**, the first large-scale resource dedicated to *procedural reasoning in biological experimental protocols*, containing a BioProCorpus of nearly 27,000 protocols and over 550,000 structured instances, covering diverse subfields of biology.

- We design five task families (Protocol Question Answering, PQA; Step Ordering, ORD; Error Correction, ERR; Protocol Generation, GEN; and Protocol Reasoning, REA) that systematically capture the unique challenges of real-world protocols, from strict step ordering to quantitative, and safety-critical reasoning.

- We establish a comprehensive evaluation with novel domain-specific metrics to conduct a fine-grained assessment of 10 LLMs, revealing systematic weaknesses in their ability to understand, reason about, and generate scientific procedural text.

- We further develop and evaluate **ProAgent**, a retrieval-augmented agent that leverages BioProBench to substantially improve both performance and reliability in protocol-related tasks, demonstrating the practical utility of the benchmark.

## 2 DESIGN AND CONSTRUCTION OF BIOPROBENCH

### 2.1 BIOPROCORPUS COLLECTION AND CLEANING

The foundation of BioProBench is a new, large-scale corpus comprising 26,933 full-text protocols collected from six authoritative resources: *Bio-protocol*, *Protocol Exchange*, *JOVE*, *Nature Protocols*, *Morimoto Lab*, and *Protocols.io* (details in Appendix D). The corpus spans 16 biological subfields, including Genomics, Immunology, and Synthetic Biology. This broad distribution, as illustrated in Figures 1(b) and (c), reflects the interdisciplinary nature of modern life science and ensures the benchmark's generalizability across diverse experimental contexts.

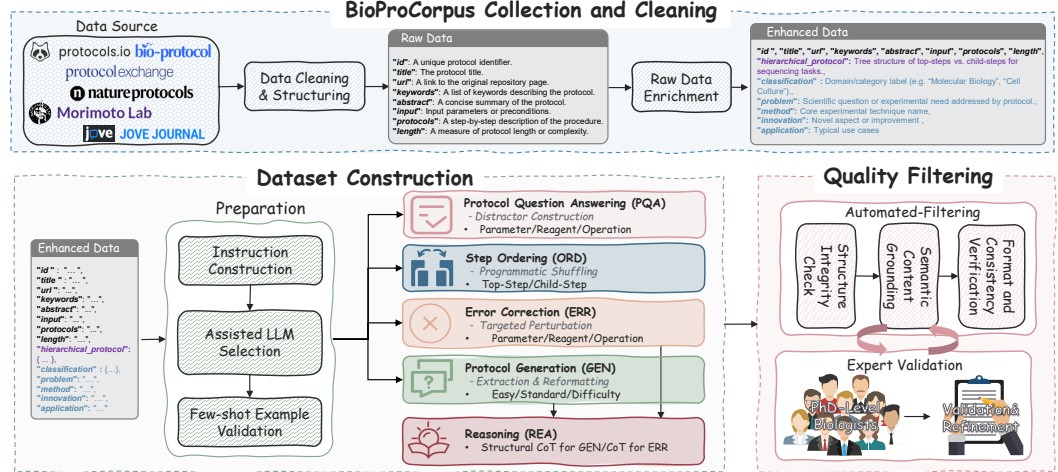

Figure 2: The construction pipeline for the **BioProBench**. The process comprises three core stages. First, a structured BioProCorpus is created by collecting, cleaning, and enriching raw scientific protocols. Second, five distinct tasks are constructed from this corpus. Finally, the benchmark goes through the quality filtering process, combining automated filtering with validation by experts.

To transform this corpus into a structured foundation, we employed a two-stage processing pipeline. The first stage involved data cleaning, including deduplication and the removal of formatting artifacts (*e.g.*, HTML tags) via regular expressions. The second stage performed structured extraction of key elements such as protocol titles, keywords, and operational steps. To preserve the inherent procedural hierarchy crucial for this work, we applied parsing rules based on indentation and symbolic cues to resolve complex nested structures like sub-steps. This process yielded a high-fidelity, structured representation of each protocol, establishing a robust basis for all subsequent task formulation. The overall workflow is illustrated in Figure 2, with further details in Appendix A.

## 2.2 DATASET AND BENCHMARK CONSTRUCTION

From the BioProCorpus, we constructed a large-scale, multi-task dataset designed to probe distinct facets of procedural reasoning. This dataset is partitioned into a large training set and a rigorously curated benchmark (held-out test set). Our methodology adheres to a strict principle: **all scientific facts, procedural steps, numerical values, and ground-truth answers are extracted programmatically and directly from the human-authored source protocols.** To augment the dataset, the LLM's role was strictly confined to functioning as a highly constrained tool, for instance, generating plausible distractors or applying minimal perturbations under programmatic control. To ensure the benchmark's scientific quality, all instances designated for the test set underwent a meticulous human expert review to verify their accuracy, remove potential artifacts, and confirm their relevance. The resulting dataset comprises 556,171 structured instances, illustrated in Figure 1(d) and Figure 3. Specifically, 380,697 instances originating from publicly licensed resources are released.

**Protocol Question Answering (PQA).** This task assesses high-precision information retrieval across three dimensions: reagent dosages, parameter values, and operational instructions. To mirror real-world challenges, we construct multiple-choice questions targeting ambiguous text segments. A ground-truth answer is first extracted directly from the source text; the LLM's role is then strictly confined to generating syntactically plausible but semantically incorrect distractors around this fixed anchor. For instances in the benchmark, every distractor was rigorously verified by experts to ensure it represents a meaningful and non-trivial challenge. Details are provided in Appendix C.1.

**Step Ordering (ORD).** This task evaluates the understanding of procedural hierarchy and causal dependencies. We designed two formats targeting global (Top-Step) and local (Child-Step) coherence. Instances are constructed by programmatically shuffling original protocol steps according to predefined rules. This deterministic process ensures the task's integrity as an objective measure of procedural logic reconstruction. The format is described in Appendix C.2.

**Error Correction (ERR).** This task assesses the critical ability to identify steps that pose safety or validity risks. Instances were created by introducing subtle errors into correct protocol steps. The

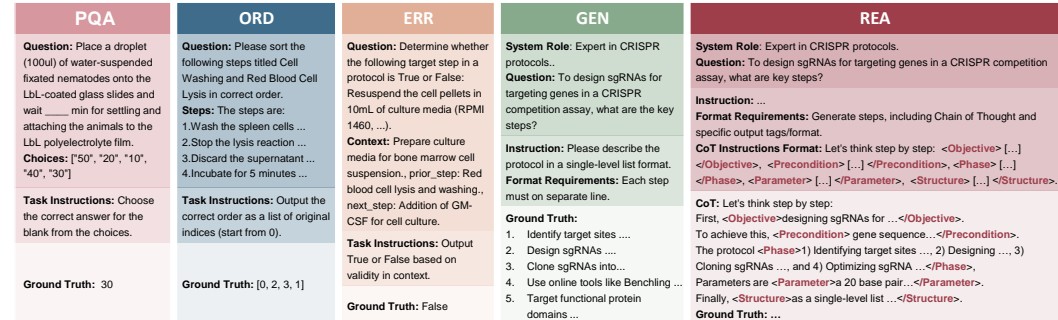

Figure 3: Representative samples for each task in the **BioProBench** benchmark.

LLM's function was limited to performing targeted, minimal perturbations (*e.g.*, altering a numerical value) under strict constraints, where the scientific context and the nature of the error were entirely predefined by the original protocol. Details on the format are in Appendix C.3.

**Protocol Generation (GEN).** This task evaluates the synthesis of coherent, long-form procedures. The process is fundamentally a task of instruction-following and content assembly, not de novo creation. Based on key information extracted from the source text, the LLM is prompted to structure a complete protocol. This requires organizing and connecting steps that are all explicitly present in the provided context, with difficulty defined by the complexity of the required procedural structure. The format is detailed in Appendix C.4.

**Protocol Reasoning (REA).** This task extends the GEN and ERR formats to include structured Chain-of-Thought (CoT) prompting (Wei et al., 2022), making the model's reasoning process explicit. For the GEN task, a template guides the model to first outline the experiment's objective, preconditions, and phases before generating the final steps. This allows for a fine-grained evaluation of whether the model's internal plan is coherent and scientifically sound, as detailed in Appendix C.5.

### 2.3 Quality Filtering

To ensure the scientific fidelity and reliability of the entire dataset, we implemented a multi-stage quality assurance process. This process combines a scalable, automated filtering pipeline with large-scale manual validation by domain experts. The three-stage automated pipeline, applied to all 556,171 instances, includes:

1. Structural Integrity Check: A rule-based check validates structural integrity of each instance, flagging malformed data, incorrect step indices, or mismatches between Q&A.

2. Semantic Content Grounding: A semantic similarity test filters out instances that deviate significantly from source protocol text, thereby mitigating potential content hallucination.

3. Format and Consistency Verification: A template-based check ensures adherence to task-specific formats and other logical constraints.

To validate the effectiveness, an extensive random sample of over 55,000 instances (10% of the dataset) was subjected to rigorous manual review by five PhD-level biologists. For the held-out test set served as the benchmark, every instance was scrutinized. The expert review focused on scientific validity, accuracy of terminology, procedural logic, and safety constraints. Hybrid quality assurance strategy allows us to achieve the scale necessary for a comprehensive benchmark while maintaining high degree of fidelity required for scientific evaluation.

### 3 Evaluation Metrics

Evaluating the procedural and semantic correctness of scientific protocols requires metrics beyond standard lexical overlap. Generated protocols may be lexically similar to references but exhibit operational flaws, such as omitting key steps or introducing unsafe parameters. To address this shortcoming, we propose a hybrid evaluation framework that combines standard NLP metrics with two sets of domain-specific metrics designed to assess scientific content and procedural fidelity. An overview is provided in Table 1, with standard metrics detailed in Appendix F.

Table 1: Evaluation metrics for the BioProBench framework. The arrow indicates the preferred direction for each metric ($\uparrow$ for higher is better, $\downarrow$ for lower is better).

| Task | Standard Metrics | Domain-specific Metrics |
|------|------------------|-------------------------|
| PQA | *Accuracy (Acc.)$\uparrow$, Brier Score(BS) $\downarrow$, Failed$\downarrow$* | - |
| ORD | *Exact Match (EM) $\uparrow$, Kendall's Tau ($\tau$) $\uparrow$, Failed $\downarrow$* | - |
| ERR | *Accuracy (Acc.) $\uparrow$ Precision (Prec.) $\uparrow$,Recall $\uparrow$, F1 $\uparrow$* | - |
| GEN | *BLEU $\uparrow$, METEOR $\uparrow$, ROUGE-L $\uparrow$* | *Keyword Precision$\uparrow$, Keyword Recall$\uparrow$, Keyword F1 $\uparrow$, Step Recall (SR), Step Precision (SP) $\uparrow$* |
| REA | *Accuracy (Acc.) $\uparrow$, Precision (Prec.) $\uparrow$,Recall $\uparrow$, F1 $\uparrow$, Failed $\downarrow$* | *LLM-as-a-Judge Consistency (Consist.) $\uparrow$* |

**Keyword-Based Content Metrics.** The accurate inclusion of critical domain-specific terminology (*e.g.*, reagents, equipment) is paramount for the scientific validity of a biological protocol. To quantify a model's ability to generate semantically relevant content, we employ a keyword-based analysis. We use KeyBERT (Grootendorst, 2021) to extract the top $k = 64$ keywords from both the reference ($K_{ref}$) and generated ($K_{gen}$) texts. From these sets, we compute **Keyword Precision** ($P_K = \frac{|K_{ref} \cap K_{gen}|}{|K_{gen}|}$), **Keyword Recall** ($R_K = \frac{|K_{ref} \cap K_{gen}|}{|K_{ref}|}$), and **Keyword F1** ($F1_K = \frac{2 \cdot P_K \cdot R_K}{P_K + R_K}$). This provides a targeted measure of whether the core scientific entities are correctly generated.

**Embedding-Based Structural Metrics.** We introduce step-level metrics to evaluate the structural fidelity of generated protocols, a critical aspect that lexical metrics fail to capture. A usable protocol must contain all necessary steps while minimizing extraneous or redundant ones. We represent the reference protocol as a sequence of steps $S_{ref} = [s_1, \ldots, s_{n_{ref}}]$ and the generated protocol as $S_{gen} = [s'_1, \ldots, s'_{n_{gen}}]$. Each step is embedded by "all-mpnet-base-v2" SentenceTransformer model [1], and similarity is measured by cosine similarity $\mathrm{Sim}(s, s')$ in Appendix F. We then compute:

**1) Step Recall (SR)** quantifies completeness by measuring the proportion of essential reference steps that are semantically captured in the generated output, using a similarity threshold $\delta = 0.7$. This threshold was not chosen arbitrarily but was justified through extensive sensitivity and qualitative analysis (detailed in Appendix F.2), which confirmed that this value robustly distinguishes between semantically relevant and irrelevant procedural steps.

$$SR = \frac{\left| \{ s \in S_{ref} : \max_{s' \in S_{gen}} \mathrm{Sim}(s, s') \geq \delta \} \right|}{|S_{ref}|}. \tag{1}$$

**2) Step Precision (SP)** quantifies conciseness and relevance by measuring the proportion of generated steps correspond to a reference step. Higher *SP* indicates fewer spurious or irrelevant steps.

$$SP = \frac{\left| s' \in S_{gen} : \max_{s \in S_{ref}} \mathrm{Sim}(s, s') \geq \delta \right|}{|S_{gen}|}. \tag{2}$$

# 4 PROAGENT: ACTIVATING THE CORPUS FOR HIGH-FIDELITY REASONING

To demonstrate the practical utility of the BioProBench corpus and to establish a strong baseline for future research, we developed **ProAgent**. The objective of ProAgent is not to introduce a novel agent architecture, but rather to serve as a standardized validation of our central hypothesis: that grounding LLMs in a high-fidelity, procedural knowledge corpus can directly and substantially address the critical weaknesses identified by our benchmark. Its implementation leverages a robust RAG framework to transparently measure the impact of the corpus itself.

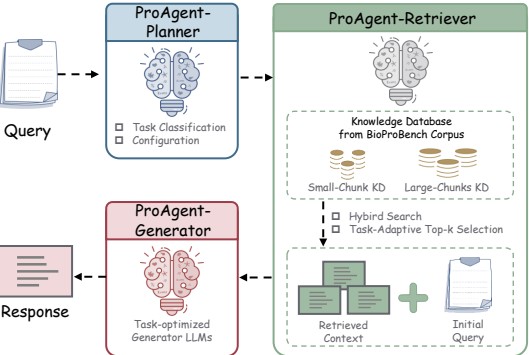

Figure 4: The architecture of ProAgent.

As illustrated in Figure 4, ProAgent operates as a task-adaptive agent that dynamically configures a specialized workflow for each query. A planner first classifies the input task (*e.g.*, PQA, GEN),

---

[1]https://www.sbert.net/

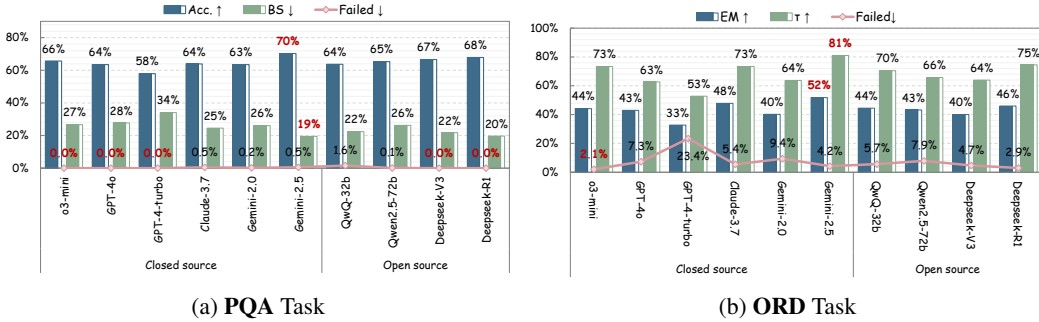

(a) **PQA** Task                          (b) **ORD** Task

Figure 5: Performance Comparison on (a) the PQA Task, measured by Accuracy (Acc) and Brier Score (BS), and (b) the ORD Task, measured by Exact Match (EM) and Kendall's Tau ($\tau$). The best value for the primary metric in each task is highlighted in red.

a decision that subsequently guides a task-adaptive retriever. This retriever selects from knowledge bases with varying chunk granularities, employing concise chunks for fact-centric tasks (PQA, ERR) and more contextual chunks for procedural ones (ORD, GEN). It utilizes a hybrid search algorithm combining semantic and lexical signals to ensure optimal context retrieval. Finally, the retrieved context is synthesized by a generator, which utilizes the best-performing baseline LLM identified for each specific task. This modular, adaptive design ensures an optimally configured response for any given procedural challenge. Detailed hyperparameters are provided in Appendix E.3.

## 5 EXPERIMENTS

We evaluated several Large Language Models (LLMs) on the BioProBench benchmark to establish a comprehensive view of their capabilities on procedural biological tasks. Our evaluation included 10 state-of-the-art general-purpose models, spanning both closed-source and open-source categories. Additionally, we assessed several smaller-scale, domain-specific Bio-LLMs; due to their distinct architectural and training paradigms, a detailed analysis of their performance is provided in Appendix G.6. Our evaluation of Bio-LLMs indicates they struggle with the procedural reasoning in BioProBench, suggesting this capability is an emergent property of larger models not fully captured by domain adaptation at this scale. A granular analysis further reveals nuanced performance, with models exhibiting unique strengths in specific sub-domains, which underscores BioProBench's utility for fine-grained capability assessment. For the main evaluation, we used a held-out test set of approximately 1,000 instances per task. Full implementation details are provided in Appendix E.

### 5.1 LLM PERFORMANCE ON THE BIOPROBENCH BENCHMARK

**Protocol Question Answering (PQA).** This task revealed varied comprehension abilities (Figure 5(a)). The top-performing closed-source model, Gemini-2.5-pro-exp, achieved the highest accuracy (70.27%), while leading open-source models like Deepseek-R1 (67.83%) demonstrated highly competitive performance. We observed a strong correlation between accuracy and confidence calibration (**BS**), where top performers also exhibited the most reliable confidence estimates. Further analysis, detailed in Appendix G.2, reveals a consistent trend across models: they perform best on qualitative tasks like identifying procedural steps ('Operation' questions) but struggle with quantitative details ('Parameter' questions), suggesting a key limitation in precise numerical comprehension.

**Step Ordering (ORD).** The ORD task proved challenging for all models, exposing significant limitations in reasoning about procedural dependencies (Figure 5(b)). Performance on ***Exact Match (EM)*** was low, with the best model achieving only 51.80%, indicating a failure to reconstruct the correct global sequence in nearly half of cases. However, higher ***Kendall's Tau ($\tau$)*** scores (0.81 for the top model) suggest most models possess a reasonable grasp of local, pairwise step relationships. A high failure rate across models, primarily due to missing or extra steps in the output, reveals a fundamental struggle with maintaining structural constraints (details in Appendix G.3). This difficulty is exacerbated as the sequence length increases, leading to a sharp decline in performance and a higher failure rate, as shown for the best-performing model in Appendix G.3.

Table 2: Performance Comparison on **ERR** Task. The best value is highlighted in blue, and the runner-up value is highlighted in light blue.

| Type | Model | Acc.↑ | Prec.↑ | Recall↑ | F1↑ | Failed↓ |
|------|-------|-------|--------|---------|-----|---------|
| Closed source | o3-mini (OpenAI, 2025) | 0.6233 | 0.8443 | 0.2993 | 0.4420 | 0.00% |
| | GPT-4o (Achiam et al., 2023) | 0.6267 | 0.7500 | 0.3763 | 0.5011 | 0.00% |
| | GPT-4-turbo (OpenAI, 2023) | 0.5617 | 0.8000 | 0.1605 | 0.2674 | 0.00% |
| | Claude-3-7-sonnet (Anthropic, 2025) | 0.6093 | 0.7363 | 0.3367 | 0.4621 | 0.17% |
| | Gemini-2.0-flash (Google AI Blog, 2024) | 0.5867 | 0.7090 | 0.2893 | 0.4109 | 0.00% |
| | Gemini-2.5-pro-exp (Google AI Blog, 2025) | 0.6483 | 0.7009 | 0.5134 | 0.5927 | 0.00% |
| Open source | QwQ-32b (Qwen Team, 2025) | 0.6300 | 0.6552 | 0.5435 | 0.5941 | 0.00% |
| | Qwen2.5-72b-instruct (Qwen Team, 2024) | 0.5917 | 0.7500 | 0.2709 | 0.3980 | 0.00% |
| | Deepseek-V3 (Liu et al., 2024a) | 0.5858 | 0.7306 | 0.2676 | 0.3917 | 0.00% |
| | Deepseek-R1 (Guo et al., 2025) | 0.6292 | 0.6197 | 0.6622 | 0.6403 | 0.00% |

Table 3: Performance Comparison on **REA-ERR** Task. The best value is highlighted in blue, and the runner-up value is highlighted in light blue.

| Type | Model | Acc. | Prec. | Recall | F1 | Failed | Consist. |
|------|-------|------|-------|--------|-----|--------|----------|
| Closed source | o3-mini (OpenAI, 2025) | 0.6505 | 0.8352 | 0.3729 | 0.5156 | 0.08% | 0.2962 |
| | GPT-4o (Achiam et al., 2023) | 0.6408 | 0.6803 | 0.5268 | 0.5938 | 0.00% | 0.2945 |
| | GPT-4-turbo (OpenAI, 2023) | 0.6033 | 0.7699 | 0.2910 | 0.4223 | 0.00% | 0.1547 |
| | Claude-3-7-sonnet (Anthropic, 2025) | 0.6508 | 0.7493 | 0.4498 | 0.5622 | 0.00% | 0.2146 |
| | Gemini-2.0-flash (Google AI Blog, 2024) | 0.6003 | 0.7542 | 0.2977 | 0.4269 | 0.33% | 0.1780 |
| | Gemini-2.5-pro-exp (Google AI Blog, 2025) | 0.6850 | 0.7273 | 0.5886 | 0.6506 | 0.00% | 0.3943 |
| Open source | QwQ-32b (Qwen Team, 2025) | 0.6421 | 0.5942 | 0.8943 | 0.7140 | 0.58% | 0.2280 |
| | Qwen2.5-72b-instruct (Qwen Team, 2024) | 0.6300 | 0.7175 | 0.4247 | 0.5336 | 0.00% | 0.2529 |
| | Deepseek-V3 (Liu et al., 2024a) | 0.6150 | 0.7906 | 0.3094 | 0.4447 | 0.00% | 0.1980 |
| | Deepseek-R1 (Guo et al., 2025) | 0.6299 | 0.5903 | 0.8353 | 0.6917 | 0.25% | 0.4526 |

**Error Correction (ERR).** Performance on this binary classification task was moderate, with accuracies ranging from 58% to 65% (Table 2). The results highlighted a distinct trade-off between *Precision* and *Recall*. Some models, like GPT-4-turbo, were highly precise (80.00%) but had very low recall (16.05%), acting as conservative error detectors. In contrast, open-source models such as Deepseek-R1 achieved a more effective balance, attaining the highest *F1*-score (64.03%) driven by strong recall (66.22%).

**Protocol Reasoning on Error Correction (REA-ERR).** To probe the model's explicit reasoning process, the REA-ERR task required models to generate a Chain-of-Thought (CoT) before providing a final answer. This allows for a deeper analysis of not just whether a model can identify an error, but why it does so. As shown in Table 3, structured CoT prompting significantly boosts performance for reasoning-optimized models; for instance, QwQ-32b's *F1*-score rose from 59.41% (on ERR) to 71.40%, and Deepseek-R1's also substantially improved.

To evaluate the validity of the generated reasoning chains, we use a *Consistency (Consist.)* metric, employing an LLM-as-a-judge. We acknowledge the potential skepticism towards LLM-based evaluation; therefore, we strictly limited its role to a semantic matching task, verifying if the model's stated error reason aligns with the ground truth, rather than an open-ended scientific evaluation. **To empirically validate this approach, we conducted a blinded human evaluation on 200 randomly sampled instances. The results showed a 94.21% agreement rate between the LLM-judge and human domain experts.** This high level of agreement provides strong support for the metric's reliability in this specific context. Crucially, the gap between models' predictive accuracy and their reasoning consistency reveals that models often reach the correct answer through invalid reasoning. More detailed explanations are in Appendix G.7.

**Protocol Generation (GEN).** The GEN task revealed a comprehensive failure across all models (Figure 6). While standard fluency metrics like *BLEU* were uniformly low (max 10.23%), and the best *METEOR* score was 24.78%, domain-specific metrics exposed more critical scientific flaws. The two most significant failures were twofold: *Step Recall (SR)* below 43%, models omitted more than half of the necessary steps, while low *Step Precision (SP)* (20%–32%) showed that their outputs were diluted with irrelevant or fabricated steps. This widespread inability to generate complete and accurate protocols highlights a profound limitation of current models.

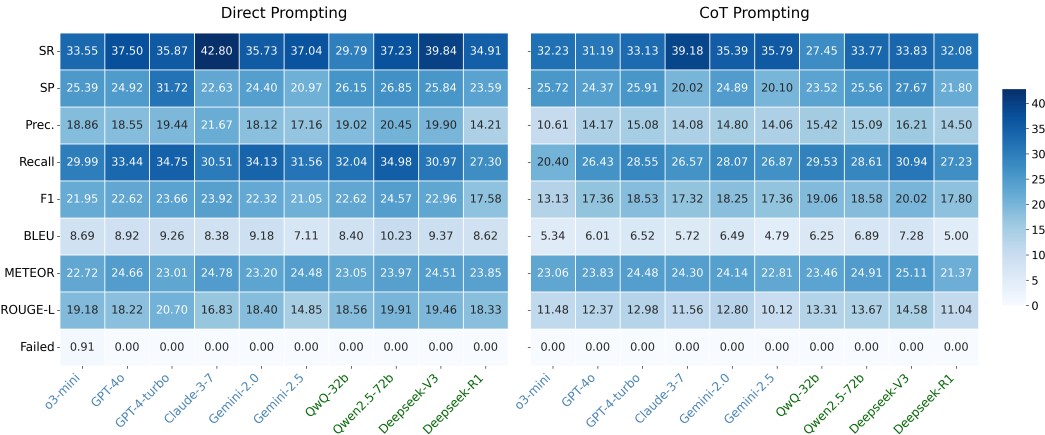

Figure 6: Comprehensive Performance Comparison on GEN Task under Direct and Zero-Shot Chain-of-Thought (CoT) Prompting.

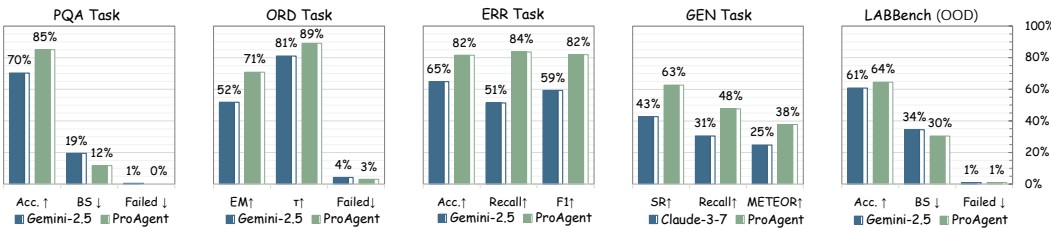

Figure 7: Comprehensive Performance Comparison on ProAgent and Gemini-2.5.

**Protocol Reasoning on Protocol Generation (REA-GEN).** Employing zero-shot CoT prompting (Kojima et al., 2022) consistently degraded performance across most models and metrics in the **GEN** task. Claude-3-7-sonnet's *SR* dropped from **0.4280** to **0.3918**. This suggests that untuned reasoning can disrupt coherent text generation. To address this, BioProBench provides structured CoT exemplars to serve as a basis for fine-tuning. The value of such guided reasoning is confirmed in our one-shot CoT experiments (Appendix G.4), which show that providing a single exemplar can mitigate performance loss and even improve certain metrics, unlike the disruptive zero-shot approach .

## 5.2 PROAGENT PERFORMANCE ANALYSIS

To validate the effectiveness of the BioProCorpus as a solution to the identified failures, we evaluated ProAgent on both our benchmark and an external dataset.

**Performance on BioProBench.** As detailed in Figure 7, grounding a capable LLM with our corpus via RAG yields substantial performance gains. In PQA, ProAgent increases accuracy by 15 points to 85.08%. In ERR, it achieves an F1-score of 81.9%, markedly improving upon the baseline's 59.27%. Most critically, for the GEN task, ProAgent raises Step Recall (SR) from 42.8% to 62.24%, directly mitigating the key failure mode of step omission. These gains strongly validate that BioProCorpus provides the necessary knowledge to enhance procedural fidelity and reduce hallucinations. However, while ProAgent marks a significant advancement, its performance is not perfect. For instance, a SR of 62.24% in the GEN task, while a 19-point improvement, still indicates that nearly 38% of necessary steps are omitted, highlighting that challenges in generating fully complete and accurate long-form protocols persist even when grounded with a high-quality corpus.

**Generalization on External Benchmarks.** To assess the broader utility of BioProCorpus, we evaluated ProAgent on 108 protocol-related questions from the LAB-Bench benchmark (Laurent et al., 2024), an out-of-distribution (OOD) task. Without RAG, the base model achieved 60% accuracy. After integrating the BioProBench RAG module, accuracy rose to 64%. This 4% absolute

improvement on an OOD task underscores the robust and transferable knowledge contained within our corpus, demonstrating its value as a foundational resource.

**Robustness and Validity Analysis.** To ensure the reliability of our evaluation, we conducted additional experiments on data contamination and metric validation. We performed a perturbation analysis on the PQA task. Results show that model performance remains stable even when numerical values in the context are modified, suggesting models rely on in-context reasoning rather than memorization (see Appendix I). We validated our embedding-based metrics (SR/SP) against an LLM-as-a-judge approach. Correlation analysis confirms that SR and SP align significantly better with expert judgment than traditional keyword-based metrics, verifying their effectiveness for procedural evaluation (see Appendix F.3).

## 6 RELATED WORKS

### 6.1 LLMS IN BIOMEDICAL DOMAIN ADAPTATION

While the rapid advancement of Large Language Models (LLMs) (Brown et al., 2020; Achiam et al., 2023; Anil et al., 2023; Team et al., 2023; Touvron et al., 2023b) has revolutionized NLP, their application in biomedicine is hindered by the domain shift between general and specialized corpora (Jahan et al., 2024). To mitigate this domain shift, three adaptation strategies have emerged: **Architecture-specific tuning**: BioBERT (Lee et al., 2020) adapts BERT through continued pre-training on PubMed. **Task-oriented specialization**: BioBART (Yuan et al., 2022) leverages BART's encoder-decoder framework for generation-heavy tasks like literature summarization, while BioGPT (Luo et al., 2022) employs GPT-style autoregressive modeling to attain state-of-the-art performance on complex reasoning benchmarks like PubMedQA (Jin et al., 2019). **Multimodal integration**: BioMedGPT (Luo et al., 2023) pioneers cross-modal alignment between medical imaging and text. However, these methods primarily address declarative knowledge found in literature or clinical narratives. They fall short when applied to experimental protocols, which are defined by procedural logic and precise operational language that eludes current models' comprehension.

### 6.2 BIOLOGICAL DATASETS AND BENCHMARKS

Existing biomedical benchmarks have evolved from question-answering (*e.g.*, BioASQ (Tsatsaronis et al., 2015) and PubMedQA (Jin et al., 2019)) to complex reasoning (*e.g.*, DiseaseQA (Jin et al., 2021) and ComprehensiveBioEval (Jahan et al., 2024)), but dedicated evaluation of procedural knowledge in protocols remains limited. However, evaluations focusing specifically on **biological experimental protocols** remian limited. For example, LAB-Bench (Laurent et al., 2024) includes many QA contents but only 135 related to protocols. BixBench (Mitchener et al., 2025) focuses on multi-step analysis of real-world datasets but does not specifically address protocol structure or procedural logic. In related clinical safety scenarios, CARDBiomedBench (Bianchi et al., 2025) evaluates neurological diagnosis and risk mitigation, and MedXpertQA (Zuo et al., 2025)) assess domain-specific reasoning but not experimental protocols. A few emerging resources target sub-tasks of protocol processing, such as error identification (BioLP-bench (Ivanov, 2024)), planning (BioPlanner (O'Donoghue et al., 2023)), or multimodal activity recognition (ProBio (Cui et al., 2023)). While **none systematically evaluate the comprehensive challenges of real-world protocols**, which require strict step ordering, causal and conditional logic, precise quantitation, and safety compliance across the key dimensions of understanding, reasoning, and generation. **BioProBench** fills this critical gap as the first large-scale benchmark explicitly dedicated to *procedural reasoning in biological protocols*, thereby complementing existing declarative benchmarks with a focus on experimental fidelity and laboratory automation.

## 7 CONCLUSION

**Summary.** In this work, we introduced BioProBench, a large-scale resource for advancing procedural reasoning in scientific AI. We presented three key assets: a foundational **BioProCorpus** of 27,000 human-authored protocols; a large **dataset** of over 550,000 structured instances for training

and evaluation; and a rigorous **benchmark** to diagnose model capabilities. Our evaluation reveals that leading LLMs have systemic weaknesses in handling causal logic, quantitative precision, and structured generation. We demonstrate that these limitations can be substantially mitigated: **ProAgent**, a RAG-based agent grounded in our corpus, achieves significant performance gains across all tasks, validating the corpus's value as a high-fidelity knowledge source. BioProBench thus provides a dual contribution: a benchmark to diagnose LLM deficiencies and a foundational resource for building more reliable scientific agents.

**Limitation & Future Work.** Key limitations include the reliance on LLMs for task structuring, which may introduce subtle model-specific artifacts, and a focus on textual protocols that omits real-world multimodal context (*e.g.*, images, videos). Future work will proceed in two directions: incorporating multimodal data, which will necessitate the development of novel evaluation frameworks to assess visual grounding and cross-modal reasoning in procedural tasks, and, building on ProAgent's success, developing protocol-specialized LLMs using the BioProBench dataset for lightweight adaptation (*e.g.*, LoRA) and advanced RAG architectures.

## ETHICS STATEMENT

The authors have read and adhered to the ICLR Code of Ethics. This work is centered on the creation of a public dataset and benchmark, and we have taken several steps to address potential ethical considerations.

**Data Sourcing and Licensing:** The BioProCorpus is constructed entirely from publicly accessible scientific protocol repositories. We have meticulously documented the source and license of every protocol used. As detailed in Appendix D, we strictly adhere to the licensing terms of each source. The publicly released portion of our benchmark (approximately 380,000 instances) is derived exclusively from sources with open licenses (*e.g.*, CC BY 4.0), ensuring full compliance and legal redistribution. Data derived from sources with restrictive licenses are not redistributed and are used only for internal validation.

**Potential for Harmful Insights or Applications:** Our research aims to improve the safety and reliability of AI in automated scientific experimentation. By creating a benchmark that specifically probes for failures in procedural logic, quantitative accuracy, and safety-critical reasoning (*e.g.*, the ERR task), our goal is to identify and mitigate the risks of LLMs generating incorrect or unsafe experimental protocols. We believe the primary application of this work is beneficial, contributing to more robust and trustworthy AI for science.

**Dataset Integrity:** To ensure the scientific validity and minimize potential artifacts introduced by LLM assistance in task formatting, all benchmark instances underwent a rigorous quality filtering pipeline, including programmatic checks and manual validation by five PhD-level domain experts, as described in Section 2.3.

## REPRODUCIBILITY STATEMENT

We are committed to ensuring the reproducibility of our work. To facilitate this, we provide comprehensive details of our methodology, data, and experiments.

**Code and Data:** The complete BioProBench benchmark (publicly licensed portion), and the source code for our evaluation pipeline and the ProAgent implementation are available at the following anonymous repository: `https://anonymous.4open.science/r/Anonymization-112358`.

**Methodology and Implementation Details:**

- The data collection, cleaning, structuring, and quality filtering processes are thoroughly described in Section 2 and detailed in Appendix A, B, and C, with representative examples for each task.
- All evaluation metrics are formally defined in Section 3, with further details in Appendix F. The full list of evaluated models, their versions, and the experimental setup are provided in Appendix E. The specific prompts used for each task are available in Appendix E.4.
- The architecture and implementation details of ProAgent are described in Section 4 and Appendix E.3, providing a clear basis for reproducing our baseline results.

These resources should provide the necessary components for the community to verify our results and build upon this work.

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

## A    CORPUS PROCESSING DETAILS

### A.1    RAW DATA PROCESSING

The raw Markdown files underwent a comprehensive multi-stage cleaning pipeline. We first dedu-plicated entries to ensure uniqueness and remove redundancies. Subsequently, we normalized their structure by converting varying formats of headings, lists, and code blocks into a consistent rep-resentation. Protocols exhibiting missing or malformed sections, or those failing initial parsing checks, were filtered out during this process to maintain data quality. After this cleaning and filter-ing, each remaining protocol was serialized as a structured JSON object, containing the following key fields: `"id"`: A unique protocol identifier; `"title"`: The protocol's official title; `"url"`: A direct link to the original protocol page on its repository; `"keywords"`: A curated list of keywords summarizing the protocol's content; `"abstract"`: A concise summary or overview of the proto-col; `"input"`: Input parameters, required materials, or preconditions; `"protocols"`: The core step-by-step description of the experimental procedure; `"length"`: Protocol length.

This structured representation facilitates efficient downstream data access, querying, and processing, serving as the foundational data layer for subsequent task construction within **BioProBench**.

### A.2    HIERARCHICAL ENRICHMENT AND ANNOTATION

To capture the intricate multi-level structure inherent in biological protocols, beyond just linear steps, we applied Large Language Models for hierarchical parsing and extraction of key semantic descrip-tors. Specifically, we leveraged LLaMA (Touvron et al., 2023a) and Deepseek-v3 (Liu et al., 2024b) models for these tasks. Following this enrichment process, we augmented the original JSON data objects with additional structured fields: `"hierarchical_protocol"`: A nested dictionary or-ganizing the linearized protocol steps into a tree-like hierarchy of up to four levels; `"problem"`, `"method"`, `"innovation"`, `"application"`, `"description"`: Automatically extracted key descriptors designed to succinctly capture the scientific context, methodology, novel aspects, po-tential applications, and a general overview of the protocol; `"classification"`: A dictionary providing domain classification of the protocol, including fields such as `"primary_domain"`, `"all_domains"` (listing all relevant biological domains), and a numerical `"confidence"` score indicating the model's certainty for the predicted classification. This hierarchical and semantic en-richment is crucial for generating sophisticated, task-specific prompts that effectively leverage not only the raw textual content of the protocols but also their underlying structural organization, de-pendencies, and key scientific attributes. The enhanced data structure example is as follows:

---

**Example of Enhanced Data**

```
"id": "Protocol.io-11",
"title": "Protein Immunoprecipitation (IP) from HEK293 Cells or iNeurons",
"url": "https://www.protocols.io/view/protein-immunoprecipitation-ip-from-hek293-cells-
    o-dpyj5pun",
"keywords": ["Protein Immunoprecipitation","HEK293 Cells",...],
"abstract": "This protocol described the methods used to isolate...",
"input": "Template DNA, primers, dNTPs, Taq polymerase...",
"protocols": "1. Protein Extraction from HEK293 Cells or iNeurons... 2.
    Immunoprecipitation...",
"length": 1471,
"hierarchical_protocol": {"1": {"title": "Protein Extraction from HEK293 Cells or
    iNeurons"},..."2": {"title": "Immunoprecipitation"},...},
"classification": {
    "primary_domain": "Genomics Technologies",
    "all_domains": ["Genomics Technologies", "Bioinformatics Methods", "Molecular
        Biology Techniques"], "confidence": 0.95},
"description": "SHARE-seq protocol v2.2 addresses the need for a scalable and efficient
    method...",
"problem": "Need for an advanced and scalable method to simultaneously profile
    chromatin accessibility and gene expression in single cells...",
"method": "An updated version of SHARE-seq, leveraging combinatorial indexing and split
    -pool barcoding ...",
"innovation": "Improved scalability, reproducibility, and efficiency for large-scale
    data production...",
"application": "Used by the Epigenomics Platform and Gene Regulation Observatory at the
    Broad Institute for the IGVF project..."
```

---

## B  BENCHMARK CONSTRUCTION DETAILS

### B.1  DETAILS OF THE CONSTRAINED LLM-ASSISTED STRUCTURING PROCESS

Our task generation pipeline was meticulously designed to ground all content in human-authored protocols while minimizing creative input from Large Language Models (LLMs). The pipeline, illustrated in Figure 2 in the following key stages.

We first designed standardized, parameterized instruction templates for each task. These templates define the task's objective, input/output format, and difficulty constraints. They contain slots where specific content from the source protocols (*e.g.*, a list of steps, a sentence with a numerical parameter) is programmatically inserted. The template itself dictates the structure of the task, not the LLM.

We selected specific models for their proficiency in reliable, low-level text manipulation tasks (*e.g.*, Deepseek-V2 (Liu et al., 2024a) for QA distractor generation, Deepseek-V3 (Liu et al., 2024b) for ERR perturbations). The selection was based on small-scale experiments focused on instruction-following fidelity rather than generative creativity.

For each protocol, our system extracts relevant content and inserts it into the corresponding instruction template to create a highly specific prompt. The LLM's role is strictly confined to executing the instruction within this prompt. For example, the prompt explicitly provides the correct answer and instructs the model only to generate four other plausible-sounding distractors for PQA. In this capacity, the LLM functions as a constrained string manipulation tool, not a knowledge source.

A small set of manually curated, high-quality examples for each task was used as a reference. Generated instances were automatically compared against these examples for structural and format consistency. Any outputs that deviated from the required format were discarded before entering the main quality filtering pipeline.

This human-in-the-loop, programmatically-guided process ensures that the LLM's role is confined to that of a highly constrained assistant for task re-formatting, thereby preserving the scientific integrity of the benchmark.

## C  BENCHMARK EXAMPLES

Below we present one representative example for each of the five core tasks. Additional examples are available in our public repository.

### C.1  PROTOCOL QUESTION ANSWERING (PQA)

---
**Examples of PQA task instances**

```
{
    "question": "Inoculate a 50˜mL culture in a 250˜mL flask with an initial OD_600nm
        of approximately ____.",
    "answer": "0.05",
    "choices": ["0.01", "0.10", "0.20", "0.50", "0.05"],
    "type": "parameter"
},
{
    "question": "Resuspend the cell pellet in 2˜mL of ____ solution or alternative
        solutions based on bacterial strain requirements.",
    "answer": "2% NaCl",
    "choices": ["1% NaOH", "3% glycerol", "0.25% yeast extract", "5.5% NaClO", "2% NaCl
        "],
    "type": "reagent"
},
{
    "question": "____ to homogenize the mixture after incubation.",
    "answer": "Gently mix",
    "choices": ["Vortex vigorously", "Centrifuge", "Boil", "Freeze", "Gently mix"],
    "type": "operation"
}
```
---

## C.2 STEP ORDERING (ORD)

---

**Example of ORD task instance**

```
"question": "Please sort the following steps titled 'Bacterial Growth Under
    Accumulation and No-Accumulation Conditions' in the correct order.",
"wrong_steps": ["Plate Preparation",
                "Overnight Culture",
                "Growth Monitoring",
                "Main Culture"],
"correct_steps": ["Plate Preparation",
                  "Overnight Culture",
                  "Main Culture",
                  "Growth Monitoring"]
```

---

## C.3 ERROR CORRECTION (ERR)

---

**Example of ERR task instance**

```
"context": {
       "purpose": "Sterilization of seeds to remove surface contaminants using sodium
            hypochlorite.",
       "prior_step": "1.1 Place transgenic Arabidopsis seeds in a 1.5mL tube.",
       "next_step": "1.2.2 Vigorously mix the contents of the tube using a vortex mixer
            ."
       },
"corrupted_text": "Mix 860\muL of sterile deionized water and 14\muL of 5% sodium
    hypochlorite in a 1.5mL tube.",
"corrected_text": "Mix 860\muL of sterile deionized water and 140\muL of 5% sodium
    hypochlorite in a 1.5mL tube.",
"is_correct": false,
"type": "parameter",
"error_description": "Reduced sodium hypochlorite volume may lead to insufficient
    sterilization.",
```

---

## C.4 PROTOCOL GENERATION (GEN)

---

**Example of GEN task instance**

```
"system_prompt": "As a specialist in mouse model protocols, provide clear step-by-step
    instructions for experimental procedures.",
"instruction": "Please describe the protocol in a flat list format (using only 1., 2.,
    3. numbers). Include only the steps, not a rationale or materials list. Use
    concise language and maintain a chronological order.",
"input": "To prepare a mouse model of LPS-induced pulpitis, standardized recipes,
    autoclaving, and anesthesia protocols have been completed. How to perform the
    Modeling Surgery?",
"output": [
  "1. Weigh the C57BL/6 mouse using a beaker and an electronic scale.",
  "2. Administer 1.2% tribromoethanol via intraperitoneal injection.",
  "3. Pinch the mouse's toe to check its reaction.",
  "4. Secure the mouse to the surgical pad with medical adhesive tape.",
  "5. Open the mouse's oral cavity with curved ophthalmic forceps.",
  "6. Use dressing forceps to expand the oral cavity and protect the tongue.",
  "7. Access the maxillary first molar with a diamond bur and expand the pulp exposure
      .",
  "8. Aspirate 0.1mL of 0.9% NaCl solution with a 1mL injection syringe.",
  "9. Flush out any dentin debris from the medullary foramen.",
  "10. Saturate a small cotton ball in the LPS-Pg solution.",
  "11. Keep the saturated cotton ball on the exposed pulp for 5 minutes.",
  "12. Remove the cotton ball and restore the tooth with Esthet-X flow.",
  "13. Loosen the mouse's restraints and return it to the cage after it awakens."
]
```

---

## C.5 PROTOCOL REASONING (REA)

In the REA task, we employ structured Chain-of-Thought (CoT) templates for both **Protocol Generation (GEN)** and **Error Correction (ERR)** scenarios. These templates guide the model through a clear, step-by-step reasoning process, covering experimental objectives, conditions, procedural phases and risk considerations, before it produces the final output.

For the **GEN** task, we use the following fixed format:

> Let's think step by step:
> First, <Objective> specify the core objective </Objective>;
> To achieve this, <Precondition> list existing conditions or prerequisites </Precondition>;
> The protocol must proceed as <Phase> divide the procedure into logical phases </Phase>;
> Critical parameters are <Parameter> identify key variables or settings </Parameter>;
> Finally, <Structure> define output format and hierarchy </Structure>.

In this template, *<Objective>* is used to focus on the core objectives of the experiment; *<Precondition>* clarifies the starting conditions and prerequisites; *<Phase>* divides the overall process into several logical stages; *<Parameter>* lists key variables or indicators; and *<Structure>* constrains the format and hierarchy of the final output. The model first fills in the corresponding content in each label block to form a complete framework of ideas, and then generates coherent step text based on the framework. The CoT example of GEN task is as flowing:

---

**Example of REA-GEN task instance**

```
...
"cot": "Let's think step by step:
    First, <Objective>generate, culture, and cryopreserve primitive endoderm stem cells
        (PrESCs) from mouse blastocysts to establish fully potent PrESCs co-
        expressing pluripotency and endoderm markers</Objective>.
    To achieve this, <Precondition>the following materials are required: MEF feeder
        cells, defined serum-free medium containing FGF4, heparin, CHIR99021, PDGF-AA,
         TrypLE Select, CELLBANKER 1 plus, AK02N medium, and all necessary reagents
        and equipment such as culture dishes, pipettes, centrifuge tubes, and
        cryotubes</Precondition>.
    The protocol must proceed as <Phase>1) Mitotic inactivation of MEF, 2) Mouse mating
         and blastocyst recovery, 3) MEF feeder preparation, 4) Blastocyst seeding, 5)
         PrESC derivation and culture, 6) Cryopreservation, and 7) Gene expression
        analysis</Phase>,
    where critical parameters are <Parameter>MEF density (5x10^6 cells/plate),
        Mitomycin C concentration (5 \mu g/mL), culture conditions (37 \circ C, 5% CO2
        ), TrypLE Select incubation time (5 min), and cryopreservation solution (
        CELLBANKER 1 plus)</Parameter>.
    Finally, <Structure>the protocol is structured into hierarchical operational steps
        with numeric identifiers, using imperative verbs, maintaining consistent tense
        , ensuring reagent continuity across steps, and excluding theoretical
        explanations to focus on detailed experimental procedures</Structure>."
```

---

For the **ERR** task, CoT is:

---

**Example of REA-ERR task instance**

```
...
"cot": "Parameter Error: Reduced sodium hypochlorite volume may lead to insufficient
    sterilization."
```

---

# D DATA LICENSING AND AVAILABILITY

We collected a total of 26,933 raw protocols, from which we generated approximately 556,121 downstream task instances. These source protocols were obtained from six distinct repositories, each governed by specific licensing terms. To ensure strict compliance with these terms, we have segregated the derived task instances based on the licensing of their original source protocols. Approximately 380,697 instances derived from openly licensed source content are publicly released under a Creative Commons Attribution 4.0 (CC BY 4.0) license. The remainder—originating from subscription-only or sources with unspecified or restrictive licenses—is retained exclusively for internal use and is not redistributed. Detailed licensing information for each source repository and the corresponding status of derived data is summarized below:

## D.1 OPENLY LICENSED SOURCES (DERIVED DATA RELEASED UNDER CC BY 4.0)

- **Bio-protocol** [2]: All task instances derived from Bio-protocol content are publicly released under CC BY 4.0. Original authorship and DOIs are attributed where applicable.

- **Protocol Exchange** [3]: As an open repository for community use, protocols from Protocol Exchange are licensed under terms permitting redistribution. Derived instances are published under CC BY 4.0, with appropriate citation of original entries.

- **Nature Protocols via protocols.io** [4]: We specifically sourced experimental procedures hosted on protocols.io that are designated as Nature Protocols and published under a CC BY 4.0 license. Task data derived from these specific protocols are publicly released under CC BY 4.0.

- **Protocols.io** [5]: The default licensing for user-submitted protocols on protocols.io is CC BY 4.0, which permits redistribution and adaptation with proper attribution to protocols.io. Task examples derived from such protocols are made publicly available under CC BY 4.0.

## D.2 SOURCES WITH RESTRICTIVE OR UNSPECIFIED LICENSES (DERIVED DATA RETAINED INTERNALLY)

- **JoVE (Journal of Visualized Experiments)** [6]: JoVE's protocols are proprietary content protected under subscription/license agreements. Copyright is held by JoVE and the authors, and bulk reuse requires explicit permission. Although content from JoVE informed aspects of our task design and methodology development, **no raw JoVE protocol text or task instances derived solely from JoVE protocols are publicly released.**

- **Morimoto Lab** [7]: The Morimoto Lab website does not explicitly specify an open license, implying standard copyright protection by Northwestern University and the authors. Consequently, **no Morimoto Lab source text or directly derived task instances are included in the public dataset.**

By segregating openly licensed and privately retained content, BioProBench rigorously honors the legal terms associated with each source while maximizing the portion of the benchmark that can be freely shared. Users accessing the public dataset are welcome to utilize and adapt the approximately 380K CC BY 4.0-licensed instances without restriction, in accordance with the license terms. The remaining instances—derived from proprietary or restrictively licensed protocols (approximately 176K instances)—are reserved strictly for internal verification, development, and benchmarking purposes and are not redistributed.

# E IMPLEMENTATION DETAILS

---

[2]https://bio-protocol.org/en
[3]https://protocolexchange.researchsquare.com/
[4]https://www.nature.com/nprot/
[5]https://www.protocols.io/
[6]https://www.jove.com/
[7]https://www.morimotolab.org/

### E.1 LLMs USED IN THE WORK

The Large Language Models (LLMs) evaluated in this study were selected to represent a diverse range of architectures and training paradigms. They are categorized into three groups:

- General-Purpose Proprietary LLMs: This group includes o3-mini (OpenAI, 2025), GPT-4o (version gpt-4o-2024-11-20) (OpenAI, 2024), GPT-4-turbo (OpenAI, 2023), claude-3.7-sonnet (version claude-3.7-sonnet-20250219) (Anthropic, 2025), gemini-2.0-flash (Google AI Blog, 2024), and gemini-2.5-pro-exp (version gemini-2.5-pro-exp-03-25) (Google AI Blog, 2025). These models represent the state-of-the-art in large commercial systems and serve as high-performance baselines.

- Leading Open-Source LLMs: This category comprises Deepseek-R1 (Guo et al., 2025), DeepSeek-V3 (version deepseek-v3-0324) (Liu et al., 2024b), QwQ 32B (Qwen Team, 2025), and qwen-2.5-72b-instruct (Qwen Team, 2024). These models were chosen as powerful, publicly accessible alternatives that reflect significant advancements in open-source development.

- Domain-Specific Bio-LLMs: We included BioMedGPT-10B (utilizing the BioMedGPT-LM-7B checkpoint) (Luo et al., 2023), BioMistral-7B (Labrak et al., 2024), and BioMistral-7B-DARE (Labrak et al., 2024) to assess the impact of domain-specific pre-training on biomedical corpora. BioMistral-7B-DARE was specifically selected for its enhanced performance on biomedical tasks.

Our evaluation methodology was standardized to ensure reproducibility and fair comparison. For all proprietary and open-source models accessed via APIs, we used the official endpoints with their default $generation_{c}onfig$ settings. This approach reflects a typical "out-of-the-box" usage scenario and minimizes experimenter-introduced bias from model-specific hyperparameter tuning. The domain-specific Bio-LLMs were evaluated locally using Hugging Face Transformers on a single NVIDIA A6000 GPU, adhering to the precision types (bfloat16 or float16) and default generation parameters recommended by the developers.

### E.2 EVALUATION DATASET CONSTRUCTION

The final evaluation dataset was curated from our comprehensive pool of generated task instances through a multi-stage, stratified sampling strategy designed to ensure statistical robustness.

The initial data pool was aggregated from six source repositories. To prioritize accessibility and reproducibility, instances derived from sources with restrictive licenses (JOVE and Morimoto Lab) were excluded. From the remaining openly licensed sources, we performed proportional random sampling. To ensure the dataset's representativeness, we then stratified the sample to achieve a balanced distribution across task-specific categories:

- **Protocol Generation (GEN)** task: Instances were stratified by difficulty levels, maintaining an approximate ratio of 5:2:1 for easy, standard, and difficult categories, respectively.

- **Protocol Question Answering (PQA)** task: Samples were balanced across defined question types, specifically targeting reagent, parameter, and operation-related questions, to achieve an equitable 1:1:1 distribution.

- **Error Correction (ERR)** task: A balanced 1:1 ratio of instances representing True versus False statements was enforced to mitigate potential classification biases during evaluation.

- **Step Ordering (ORD)** task: Instances were sampled to establish a 1:2 ratio between those requiring the ordering of top-level protocol steps versus those involving child-level (sub-protocol) steps, reflecting different granularity levels.

Finally, a panel of biology PhD experts conducted a rigorous manual review of all sampled instances. This human-in-the-loop validation confirmed the factual accuracy, logical coherence, and relevance of each item. This meticulous curation process resulted in a high-quality evaluation set comprising **1,200** instances for ERR, **772** for GEN, **1,200** for PQA, and **1,161** for ORD.

### E.3 ProAgent Implementation Details

ProAgent was constructed using the LangGraph library. The agent operates as a conditional graph where nodes represent processing steps (Planner, Retriever, Generator) and edges represent the flow of logic based on the planner's decisions.

**Knowledge Base Construction**: The 27,000 protocols from the corpus were split into chunks. The Small-Chunk KB was generated with a character window of 2000 and an overlap of 400. The Large-Chunk KB used a window of 5000 and an overlap of 1000. **Embedding Model**: All chunks were embedded using the Qwen-3-Embedding model. **Hybrid Search Configuration**: The weighting factor for hybrid search was set to $\alpha$=0.6, balancing semantic and lexical signals (60% semantic, 40% BM25). The FAISS index was an IndexFlatL2. BM25 was implemented using the rank-bm25 library. **Retrieval Parameters**: The initial dense retrieval stage returns 60 candidate chunks. The number of final chunks (k) passed to the generator is 5 for PQA/ERR tasks and 3 for ORD/GEN tasks.

For PQA, ORD, and ERR tasks, we utilized the Gemini-2.5-pro-exp model via its official API. For the GEN task, we used the Claude-3-7-sonnet model.

### E.4 Prompts Used in Each Task

The specific prompts used for each task in our experiments are provided below.

---

**Prompts for PQA**

```
You will be given a multiple-choice question related to a biological protocol.  The
blank in the question (represented as ___) indicates where the correct choice should be
filled in.

Question:
{question}

Choices:
{choices}

Your task:

- Choose the most likely correct answer from the given choices.

- You must always select one answer, even if you are unsure.

- The selected answer must match one of the choices exactly (including case and
punctuation).

- Assign a confidence score between 0 and 100 based on your certainty.

- Output your answer wrapped exactly between the tags [ANSWER_START] and [ANSWER_END].

- The format of your response must be:  [ANSWER_START]your selected choice & your
confidence score[ANSWER_END]
```

---

**Prompts for ERR**

```
Determine whether the following target step in a protocol is True or False:

{step}

You may use the following context, which includes the purpose of the step, as well as
the preceding and following steps, to inform your decision:

{context}

Please carefully evaluate if the step is logically consistent, necessary, and accurate
in the context.  If you find anything wrong, answer False.

- Please respond with only True or False, without any additional explanation.

- Output your answer wrapped exactly between the tags [ANSWER_START] and [ANSWER_END].

- The format of your response must be:  [ANSWER_START]True or False[ANSWER_END]
```

---

## Prompts for ORD

Please sort the following steps titled {title} in the correct order.

The steps are:

{steps out of order}

- Give me the correct order of the steps as a list of their original indices (start from 0), no other words.

- Output your answer wrapped exactly between the tags [ANSWER_START] and [ANSWER_END].

- The format of your response must be: [ANSWER_START]a list of the original indices[ANSWER_END]

## Prompts for GEN

You are a specialist in {domain} protocols, skilled in experimental procedures of {domain}.

Please describe the protocol in a flat list format (using only 1., 2., 3. numbers). Include only the steps, not a rationale or materials list. Use concise language and maintain a chronological order.

Format requirements:

- Each step must be on a separate line.

Now my question is:

{input question}

## Prompts for REA-GEN

You are a specialist in {domain} protocols, skilled in experimental procedures of {domain}.

Please describe the protocol in a flat list format (using only 1., 2., 3. numbers). Include only the steps, not a rationale or materials list. Use concise language and maintain a chronological order.

Your response must be structured strictly for machine processing. It must contain two main parts in order:
1. Your Chain of Thought (CoT) process, formatted with specific XML-like tags.
2. The final detailed protocol steps, wrapped in [ANSWER_START][ANSWER_END] tags.

Please begin your response by outputting your thinking process. Follow this exact structure and include your analysis within the respective tags:

Let's think step by step:
<Objective>[Output the core objective of this protocol here]</Objective>.

To achieve this, <Precondition>[Output the necessary preconditions, materials, equipment, etc., here]</Precondition>.

The protocol must proceed as <Phase>[Output the logical division into key phases or stages here]</Phase>,

where critical parameters are <Parameter>[Output the critical parameters for each step/phase and the logic behind them here]</Parameter>.

Finally, <Structure>[Acknowledge and state the required output structure for the final steps here]</Structure>.

After outputting the complete thinking process exactly as structured above, output the final detailed protocol steps.

Format requirements for the final output steps (which must be placed between the [ANSWER_START] and [ANSWER_END] tags):

- Each step must be on a separate line.
- [ANSWER_START] [Output the detailed protocol steps here, ensuring each step is on a new line] [ANSWER_END]

Now my question is:
{input question}

---

**Prompts for REA-ERR**

```
Evaluate the validity of the following target step in a protocol.  Follow the detailed
reasoning process demonstrated in the example below to identify potential errors across
Operation, Reagent, and Parameter categories, with meticulous attention to numerical
values and their consistency with the provided context and typical practices.

--- Example Start

Example Target Step:
Mix 860µL of sterile deionized water and 14µL of 5% sodium hypochlorite in a 1.5mL
tube.

Example Context:
"purpose":  "Sterilization of seeds to remove surface contaminants using sodium
hypochlorite.",
"prior_step":  "1.1 Place transgenic Arabidopsis seeds in a 1.5mL tube.",
"next_step":  "1.2.2 Vigorously mix the contents of the tube using a vortex mixer."

Example Reasoning Process:
1.  Operation Error:  The operations (Mix) and the use of a 1.5mL tube are standard.
No obvious operational errors.

2.  Reagent Error:  The reagents are appropriate.  However, the specified volume of
5% sodium hypochlorite is 14µL, mixed with 860µL water.  This results in a very dilute
solution (0.07%).  For sterilization, typical practice suggests a final concentration
of around 0.5{1% sodium hypochlorite.  Therefore, the reagent volume is significantly
too low, which undermines effectiveness and contradicts the stated sterilization
purpose.

3.  Parameter Error:  Although explicit parameters like time and temperature are not
mentioned, the concentration of sodium hypochlorite functions as a critical parameter
in disinfection efficacy.  Here, the final concentration (0.07%) is too low to be
effective, making it a parameter error as well.

Based on the significant numerical error in both Reagent volume and the effective
concentration (parameter), the step is invalid.

Example Answer:  [ANSWER_START]False[ANSWER_END]
--- Example End

Now, evaluate the following target step using the same detailed reasoning process
demonstrated in the example above:

Evaluate the validity of the target step:
{step}

You may use the following context, which includes the purpose of the target step, as
well as the preceding and following steps, to inform your decision:
{context}

Analyze the step, paying meticulous attention to all numerical values (e.g., times,
temperatures, volumes, concentrations, speeds, durations), by reasoning through the
following three categories of potential errors.  As part of this analysis, explicitly
compare numerical values specified in the target step and consider typical laboratory
practices.

Only evaluate the correctness of the information explicitly present in the target step.
Do not make assumptions about missing details.  Focus solely on identifying errors in
what is actually stated.

The format of your final answer must be:  [ANSWER_START]True or False[ANSWER_END]
```

# F  SUPPLEMENTARY EVALUATION METRICS

## F.1  METRICS DETAILS

**PQA**  We evaluate model performance using two key metrics: ***accuracy*** to measure answer correctness and the ***Brier score*** to assess confidence calibration. To compute these metrics, we first parse the model's generated response to extract both the predicted answer and its associated confidence score.

The model's output string is processed using regular expressions to isolate content between [ANSWER_START] and [ANSWER_END] markers. The parsed content is split into the predicted answer $\hat{a}_i$ (first segment) and a confidence score $c_i$ (numeric value extracted from the final segment). Failures in parsing (e.g., malformed output) are excluded from metric calculations and reported separately as a parsing error rate. **Accuracy** is then computed as:

$$A = \frac{1}{M} \sum_{i=1}^{M} \mathbb{I}(a_i = \hat{a}_i), \tag{3}$$

where $a_i$ is the ground-truth answer, $\hat{a}_i$ is the model's prediction, and $\mathbb{I}$ is the indicator function. **Brier Score** quantifies calibration by measuring the squared deviation between confidence and accuracy:

$$B = \frac{1}{M} \sum_{i=1}^{M} (c_i - \mathbb{I}(a_i = \hat{a}_i))^2, \tag{4}$$

with $c_i \in [0, 1]$ rescaled from the parsed confidence percentage. Perfect calibration yields $B = 0$.

For robustness, we report the parsing failure rate $\rho = F/N \times 100\%$, where $F$ is the count of unparsable responses and $N$ is the total number of questions.

**ORD** We evaluate ranking performance using two metrics: **Exact Match** for absolute sequence correctness and **Kendall's Tau ($\tau$)** for pairwise order consistency. The model's output is parsed from the text between [ANSWER_START] and [ANSWER_END] markers, converted to a list of indices $\mathbf{p} = (p_1, \ldots, p_n)$ representing the predicted order of steps. These are mapped to their corresponding step contents and compared against the ground-truth sequence $\mathbf{g} = (g_1, \ldots, g_n)$. Cases with mismatched step sets (i.e., $\text{set}(\mathbf{p}) \neq \text{set}(\mathbf{g})$) are discarded and reported as parsing failures. **Exact Match** measures the proportion of perfectly predicted sequences:

$$A_{\text{EM}} = \frac{1}{M} \sum_{i=1}^{M} \mathbb{I}(\mathbf{p}_i = \mathbf{g}_i), \tag{5}$$

where $M$ is the number of valid parsed samples and $\mathbb{I}$ is the indicator function. **Kendall's Tau ($\tau$)** quantifies rank correlation by comparing concordant/discordant pairs:

$$\tau = \frac{2}{n(n-1)} \sum_{a<b} \left[ \mathbb{I}\left((p_a - p_b)(g_a - g_b) > 0\right) - \mathbb{I}\left((p_a - p_b)(g_a - g_b) < 0\right) \right], \tag{6}$$

where $n$ is the sequence length, and $p_a, g_a$ denote the ranks of element $a$ in the predicted and ground-truth orders, respectively. $\tau = 1$ indicates perfect agreement, while $\tau = -1$ implies complete inversion.

We further report the parsing failure rate $\rho = F/N \times 100\%$, where $F$ counts unparsable or set-mismatched responses and $N$ is the total samples.

**ERR** The model's error detection capability is evaluated using standard binary classification metrics. The model's output is parsed to extract the predicted boolean label (True for correct input, False for erroneous input) through pattern matching between [ANSWER_START] and [ANSWER_END] markers. Parsing failures (e.g., missing markers or invalid labels) are excluded from metric calculations. Key metrics are defined as:

- **Precision**: The proportion of correctly identified errors among all inputs flagged as erroneous by the model. High precision indicates low false alarm rate.

- **Recall**: The proportion of actual errors successfully detected by the model. High recall reflects comprehensive error coverage.

- **F1-score**: The harmonic mean of precision and recall, balancing the trade-off between false positives and false negatives.

**GEN** To comprehensively assess the performance of Large Language Models (LLMs) on our protocol text generation benchmark, we employ a multi-faceted evaluation strategy. Recognizing the limitations of relying solely on standard text generation metrics or potentially biased LLM-based judges (given the specialized domain knowledge required for protocols, which current LLMs may lack for reliable judgment), our evaluation integrates three distinct categories of metrics: Standard Text Generation Metrics, Keyword-Based Content Metrics, and Embedding-Based Structural Metrics. This approach aims to capture fluency, core semantic content, and the crucial procedural structure inherent in biological protocols.

We include established metrics primarily as a baseline assessment of overall textual fluency and surface-level semantic similarity. While metrics like **BLEU** (Papineni et al., 2002), **METEOR** (Banerjee & Lavie, 2005), and **ROUGE-L** (Lin, 2004) (which measures longest common subsequence overlap) are standard, we acknowledge their potential insufficiency for evaluating the logical coherence and step-by-step accuracy required in long-form, structured texts such as experimental protocols. They serve as a general indicator of quality but are supplemented by more domain-specific evaluations.

In the Embedding-Based Structural Metrics, we calculate the semantic similarity between a generated step $s'$ and a reference step $s$ using Cosine Similarity. Let $\mathbf{v}_s$ and $\mathbf{v}_{s'}$ denote the vector embeddings of the reference step and the generated step, respectively, produced by the sentence transformer model. The cosine similarity is defined as the dot product of the two vectors divided by the product of their Euclidean magnitudes:

$$\text{Sim}(s, s') = \frac{\mathbf{v}_s \cdot \mathbf{v}_{s'}}{\|\mathbf{v}_s\|\|\mathbf{v}_{s'}\|} = \frac{\sum_{i=1}^{n} v_{s,i} v_{s',i}}{\sqrt{\sum_{i=1}^{n} v_{s,i}^2} \sqrt{\sum_{i=1}^{n} v_{s',i}^2}} \tag{7}$$

where $n$ is the dimensionality of the embedding vectors. The resulting score ranges from -1 to 1, with 0 indicating orthogonality.

**REA** To assess the model's ability to provide accurate reasoning alongside error correction, we evaluate its performance on two fronts: the correctness of the error detection and the validity of the provided reasoning process.

The evaluation of error detection itself mirrors the section above, utilizing standard binary classification metrics: *Precision*, *Recall*, and *F1-score*.

In addition to these, we introduce a specific metric to evaluate the quality of the reasoning provided when an error is identified: *Consistency (Consist.)*. This metric measures the proportion of actual erroneous samples for which the model correctly identifies the underlying reason for the error within its reasoning process. Let $S_{\text{err}}$ be the set of actual erroneous samples in the error correction task, and $N_{\text{err}} = |S_{\text{err}}|$ be its cardinality. Let $\mathbb{I}_{\text{reason}}(\mathbf{x})$ be an indicator function that is 1 if the model's reasoning process for sample $\mathbf{x}$ correctly identifies the true cause of the error, and 0 otherwise. The Consistency is defined as:

$$\text{Consist.} = \frac{1}{N_{\text{err}}} \sum_{\mathbf{x} \in S_{\text{err}}} \mathbb{I}_{\text{reason}}(\mathbf{x}) \tag{8}$$

For $\mathbb{I}_{\text{reason}}(\mathbf{x})$ to be 1, the model must not only successfully flag the input $\mathbf{x}$ as erroneous, but its generated reasoning process must accurately reflect the ground-truth nature of the error. To determine if the model's identified error reason aligns with the ground-truth error, we employ an LLM-as-a-judge, specifically `deepseek-v3`. The use of an LLM judge is considered appropriate here due to the relatively low complexity of comparing the model's stated error cause with the predefined ground-truth error, a task for which current LLMs demonstrate sufficient competency.

### F.2 SUPPLEMENTARY ANALYSIS OF EVALUATION METHODOLOGY

This section provides a detailed analysis to validate the robustness of our proposed evaluation framework. We specifically investigate two key aspects of the embedding-based metrics (SR and SP): the sensitivity to the choice of the similarity threshold and the dependency on a specific embedding model. Furthermore, we present a granular, sub-domain level performance analysis to offer a more nuanced understanding of model capabilities.

**Sensitivity to the Similarity Threshold ($\delta$)**   The choice of the similarity threshold, $\delta$, is a critical parameter. To investigate its impact, we evaluated all models across a wide range of thresholds, $\delta \in \{0.5, 0.6, 0.7, 0.8, 0.9\}$. This analysis yielded three key insights.

1. **Identification of a Stable Ranking Zone:** The analysis revealed that while absolute scores naturally decrease as the threshold becomes stricter, the relative rankings of the models remain highly consistent within a "stable zone." This consistency was quantified using Spearman's rank correlation ($\rho$) between the model rankings at our chosen threshold ($\delta = 0.7$) and other thresholds, as shown in Table 4. The correlation is exceptionally high (often $\geq 0.8$) within the $[0.6, 0.8]$ range, demonstrating that the choice of threshold within this zone does not significantly alter our conclusions. The sharp drop at $\delta = 0.9$ for SR is due to a "floor effect", where nearly all scores collapse to zero, making rankings unreliable. This confirms that an extreme threshold is not appropriate.

Table 4: Spearman's Rank Correlation ($\rho$) of model rankings at various thresholds compared to the rankings at our selected threshold of $\delta = 0.7$.

| Metric | Corr. with $\delta = 0.5$ | Corr. with $\delta = 0.6$ | Corr. with $\delta = 0.8$ | Corr. with $\delta = 0.9$ |
|--------|--------|--------|--------|--------|
| **SR** | 0.927 | 0.927 | 0.709 | 0.218 |
| **SP** | 0.758 | 0.927 | 0.842 | 0.612 |

2. **Qualitative Justification:** Case studies further justify the choice of $\delta = 0.7$. For example, two semantically identical sentences, such as "Use a serological pipette to detach the cells." and "Detach cells using a serological pipette.", yield a cosine similarity of **0.86**. A very strict threshold of $\delta = 0.9$ would incorrectly penalize such valid paraphrasing. Conversely, two semantically unrelated sentences scored **0.55**, confirming that a threshold around 0.7 effectively distinguishes between relevant and irrelevant content.

3. **Raw Score Stability:** For full transparency, we present the raw SR and SP scores for all models within the identified stable zone ($\delta \in \{0.6, 0.7, 0.8\}$) in Table 5 and Table 6. This data reinforces the observation that relative model performance is consistent across this range.

Collectively, this evidence demonstrates that our choice of $\delta = 0.7$ is not an arbitrary selection but is a justified, representative point within a stable evaluation landscape.

Table 5: SR scores for all models evaluated at different similarity thresholds within the stable zone.

| Model | SR ($\delta = 0.6$) | SR ($\delta = 0.7$) | SR ($\delta = 0.8$) |
|-------|--------|--------|--------|
| claude-3-7-sonnet-20250219 | 0.7176 | 0.4280 | 0.1467 |
| deepseek-v3-0324 | 0.6918 | 0.3984 | 0.1317 |
| gemini-2.5-pro-exp-03-25 | 0.6747 | 0.3704 | 0.1076 |
| gpt-4-turbo | 0.6356 | 0.3587 | 0.1169 |
| qwen2.5-72b-instruct | 0.6674 | 0.3723 | 0.1282 |
| o3-mini | 0.6315 | 0.3355 | 0.1094 |
| qwq-32b | 0.5868 | 0.2979 | 0.0838 |
| deepseek-r1 | 0.6474 | 0.3491 | 0.1129 |
| gemini-2.0-flash | 0.6624 | 0.3573 | 0.1417 |
| gpt-4o-2024-11-20 | 0.6677 | 0.3785 | 0.1180 |

**Robustness to the Choice of Embedding Model**   To ensure that our results are not dependent on a single embedding model, we conducted a robustness check by re-running our entire evaluation with a different, popular embedding model: `bge-m3`. We then compared the model rankings produced by `bge-m3` with those from our original model (`all-mpnet-base-v2`) at the chosen threshold of $\delta = 0.7$. The full results are presented in Table 7.

The consistency of the rankings produced by the two embedding models was quantified using Spearman's rank correlation. The results show a remarkably high level of agreement:

- **Spearman's $\rho$ for SR** (`all-mpnet-base-v2` vs. `bge-m3`): **0.794**
- **Spearman's $\rho$ for SP** (`all-mpnet-base-v2` vs. `bge-m3`): **0.903**

Table 6: SP scores for all models evaluated at different similarity thresholds within the stable zone.

| Model | SP ($\delta = 0.6$) | SP ($\delta = 0.7$) | SP ($\delta = 0.8$) |
|---|---|---|---|
| claude-3-7-sonnet-20250219 | 0.4667 | 0.2263 | 0.0688 |
| deepseek-v3-0324 | 0.5186 | 0.2584 | 0.0777 |
| gemini-2.5-pro-exp-03-25 | 0.4877 | 0.2097 | 0.0510 |
| gpt-4-turbo | 0.5975 | 0.3172 | 0.0929 |
| qwen2.5-72b-instruct | 0.5364 | 0.2685 | 0.0785 |
| o3-mini | 0.5121 | 0.2539 | 0.0753 |
| qwq-32b | 0.5270 | 0.2615 | 0.0830 |
| deepseek-r1 | 0.4899 | 0.2359 | 0.0675 |
| gemini-2.0-flash | 0.4756 | 0.2440 | 0.0795 |
| gpt-4o-2024-11-20 | 0.5182 | 0.2492 | 0.0704 |

Table 7: SR and SP scores computed using the `bge-m3` embedding model at a threshold of $\delta = 0.7$.

| Model | SR Score | SP Score |
|---|---|---|
| claude-3-7-sonnet-20250219 | 0.5756 | 0.3105 |
| deepseek-v3-0324 | 0.5342 | 0.3495 |
| gemini-2.5-pro-exp-03-25 | 0.4883 | 0.2816 |
| gpt-4-turbo | 0.4784 | 0.4183 |
| qwen2.5-72b-instruct | 0.5071 | 0.3601 |
| o3-mini | 0.4791 | 0.3619 |
| qwq-32b | 0.4257 | 0.3605 |
| deepseek-r1 | 0.4785 | 0.3254 |
| gemini-2.0-flash | 0.5272 | 0.3462 |
| gpt-4o-2024-11-20 | 0.4952 | 0.3263 |

These high correlation coefficients provide compelling evidence that our paper's conclusions are not an artifact of a specific tool choice. The relative performance hierarchy of the LLMs is consistently captured by different semantic evaluation models, confirming the generalizability of our findings.

## F.3 Validation of Embedding-Based Structural Metrics

To validate the effectiveness of our proposed embedding-based structural metrics (Step Recall (SR) and Step Precision (SP)), we assessed their alignment with an "LLM-as-a-judge" oracle. We utilized GPT-4o as the judge to score the "procedural correctness" of generated protocols on a scale of 1-5, analyzing a balanced subset of 51 GEN task instances.

We calculated the Spearman Rank Correlation ($\rho$) between our automated metrics and the judge's scores. As shown in Table 8, our structural metrics (SR and SP) demonstrate a statistically significant positive correlation with the judge's assessment ($\rho$ ranging from 0.50 to 0.67 for top models), consistently outperforming the traditional Keyword F1 metric.

This comparison highlights a key finding: lexical overlap metrics (like Keyword F1) are often insufficient for capturing the logical integrity of long biological protocols. The stronger alignment of SR and SP with the judge confirms that our embedding-based approach provides a more robust and semantically meaningful evaluation of procedural generation.

Table 8: Validation of Embedding-Based Structural Metrics

| Model | Metric Pair | Spearman $\rho$ | P-value |
|---|---|---|---|
| | Step Recall (SR) $\leftrightarrow$ Judge | 0.6718 | 1.71e-07 |
| Claude-3.7 | Step Precision (SP) $\leftrightarrow$ Judge | 0.5474 | 5.66e-05 |
| | Keyword F1 $\leftrightarrow$ Judge | 0.4262 | 2.52e-03 |
| | Step Recall (SR) $\leftrightarrow$ Judge | 0.4992 | 2.24e-04 |
| o3-mini | Step Precision (SP) $\leftrightarrow$ Judge | 0.6013 | 3.87e-06 |
| | Keyword F1 $\leftrightarrow$ Judge | 0.4312 | 1.77e-03 |
| | Step Recall (SR) $\leftrightarrow$ Judge | 0.4947 | 2.60e-04 |
| DeepSeek-R1 | Step Precision (SP) $\leftrightarrow$ Judge | 0.4390 | 1.42e-03 |
| | Keyword F1 $\leftrightarrow$ Judge | 0.3405 | 1.55e-02 |

# G   ADDITIONAL EXPERIMENTS

## G.1   DETAILED RESULTS FOR PQA AND ORD

The detailed results across different models in PQA task are shown in Table 9 and Table 10.

Table 9: Performance Comparison on **PQA** Task. The best value is highlighted in blue, and the runner-up value is highlighted in light blue.

| Type | Model | Acc. ↑ | BS ↓ | Failed ↓ |
|---|---|---|---|---|
| Closed source | o3-mini (OpenAI, 2025) | 0.6567 | 0.2665 | 0.00% |
| | GPT-4o (Achiam et al., 2023) | 0.6350 | 0.2770 | 0.00% |
| | GPT-4-turbo (OpenAI, 2023) | 0.5792 | 0.3403 | 0.00% |
| | Claude-3-7-sonnet (Anthropic, 2025) | 0.6390 | 0.2454 | 0.50% |
| | Gemini-2.0-flash (Google AI Blog, 2024) | 0.6344 | 0.2605 | 0.17% |
| | Gemini-2.5-pro-exp (Google AI Blog, 2025) | 0.7027 | 0.1949 | 0.50% |
| Open source | QwQ-32b (Qwen Team, 2025) | 0.6367 | 0.2236 | 1.58% |
| | Qwen2.5-72b-instruct (Qwen Team, 2024) | 0.6530 | 0.2624 | 0.08% |
| | Deepseek-V3 (Liu et al., 2024a) | 0.6658 | 0.2172 | 0.00% |
| | Deepseek-R1 (Guo et al., 2025) | 0.6783 | 0.1965 | 0.00% |

Table 10: Performance Comparison on **ORD** Task. The best value is highlighted in blue, and the runner-up value is highlighted in light blue.

| Type | Model | EM↑ | τ↑ | Failed↓ |
|---|---|---|---|---|
| Closed source | o3-mini (OpenAI, 2025) | 0.4415 | 0.7333 | 2.07% |
| | GPT-4o (Achiam et al., 2023) | 0.4294 | 0.6268 | 7.32% |
| | GPT-4-turbo (OpenAI, 2023) | 0.3273 | 0.5278 | 23.43% |
| | Claude-3-7-sonnet (Anthropic, 2025) | 0.4781 | 0.7336 | 5.43% |
| | Gemini-2.0-flash (Google AI Blog, 2024) | 0.4021 | 0.6370 | 9.39% |
| | Gemini-2.5-pro-exp (Google AI Blog, 2025) | 0.5180 | 0.8104 | 4.22% |
| Open source | QwQ-32b (Qwen Team, 2025) | 0.4447 | 0.7047 | 5.68% |
| | Qwen2.5-72b-instruct (Qwen Team, 2024) | 0.4322 | 0.6572 | 7.92% |
| | Deepseek-V3 (Liu et al., 2024a) | 0.4005 | 0.6395 | 4.74% |
| | Deepseek-R1 (Guo et al., 2025) | 0.4596 | 0.7453 | 2.93% |

## G.2   PERFORMANCE ON DIFFERENT CATEGORIES OF PQA

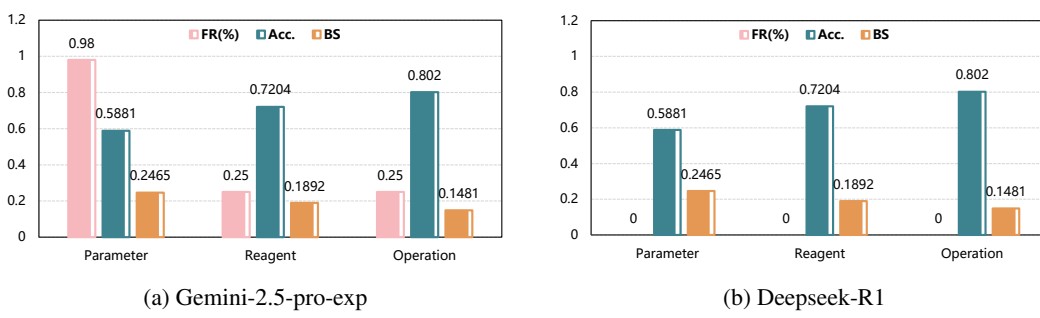

(a) Gemini-2.5-pro-exp                                    (b) Deepseek-R1

Figure 8: Performance comparison across different categories of PQA.

Performance varied significantly across different question categories, as detailed for representative models in Figure 8. Both Gemini-2.5-pro-exp (Team et al., 2023) and Deepseek-R1 (Guo et al., 2025) consistently achieved their highest accuracy and best calibration on "Operation" questions (identifying procedural steps), followed by "Reagent" questions, and performed worst on "Paramete" questions (extracting specific numerical values or conditions). This suggests that current LLMs are more adept at processing qualitative aspects of protocols (actions, sequences) than quantitative details (precise values, measurements), indicating potential limitations in handling numerical or fine-grained conditional information within scientific procedures.

## G.3 ANALYSIS OF FAILURE TYPES ON ORD TASK

A critical issue across models was the Failure Rate—the inability to produce a parsable output conforming to task requirements. This brittleness varied greatly, with some models exhibiting alarmingly high *Failed*, such as GPT-4-turbo (23.43%) and the open-source QwQ-32b (22.39%). Even top performers like Gemini-2.5-pro had a non-trivial *Failed* (4.22%). To understand the nature of these failures, we analyzed GPT-4-turbo's errors (Table 11). The analysis revealed that the majority of its 272 failures stemmed from fundamental structural issues: failing to include all required steps ("Missing steps", 58.46%) or including extra steps (32.35%). Less frequent errors involved invalid formatting or duplicated steps. These patterns indicate that beyond sequential reasoning difficulties, models struggle with basic task constraints like outputting the correct set of elements, especially under the cognitive load of permutation.

Table 11: Analysis of Failure Types for GPT-4-turbo on **ORD** Task.

| Failure Type | Count | Percentage of Failures |
|---|---|---|
| Missing steps | **159** | **58.46%** |
| Extra steps | 88 | 32.35% |
| Invalid format | 24 | 8.82% |
| Repeated steps | 1 | 0.37% |
| Total Failures | 272 | 100.00% |

Briefly comparing model types, the closed-source Gemini-2.5-pro led overall, yet open-source models like QwQ-32b and Deepseek-R1 showed competitive Tau scores, indicating potential in pairwise ordering (though QwQ-32b's high *Failed* is a caveat). Strikingly, biomedical-specific models (BioMedGPT, BioMistral variants) performed drastically worse than general-purpose ones on all metrics, including extremely high *Failed* (33%-91%), suggesting domain specialization did not aid, and perhaps hindered, performance on this procedural task.

The impact of sequence length was examined using the best model, Gemini-2.5-pro (Table 12). As the number of steps increased, *EM* accuracy plummeted (65.8% for 3-5 steps vs. 13.0% for 12+ steps), confirming the difficulty of maintaining global coherence. *Kendall's* $\tau$ remained more robust, peaking for moderately long sequences, further supporting the observation that local pairwise understanding is stronger than global sequence construction. Critically, the *Failed* escalated sharply with sequence length (1.5% for 3-5 steps vs. 16.3% for 12+ steps), showing that longer sequences exacerbate both reasoning challenges and output brittleness.

Table 12: Performance of Gemini-2.5-pro-exp across different step numbers.

| Step Num | *EM* | $\tau$ | *Failed(%)* | $\mathbf{N}_{total}$ |
|---|---|---|---|---|
| 3–5 | 0.6583 | 0.7729 | 1.54 | 648 |
| 6–8 | 0.4079 | 0.7972 | 4.81 | 291 |
| 9–11 | 0.2750 | 0.8348 | 7.69 | 130 |
| 12+ | 0.1299 | 0.8208 | 16.30 | 92 |
| Overall | 0.5180 | 0.8104 | 4.22 | 1161 |

## G.4 ONE-SHOT CoT PROMPTING ON GEN TASK

Our initial findings, detailed in Figure 6, presented results for models under both direct prompting and a zero-shot CoT approach. To further investigate the impact of CoT prompting, we then introduced a one-shot CoT condition, providing a single exemplar of the desired reasoning process before generation. The performance under this one-shot condition, detailed in Table.13, offers a more nuanced perspective compared to the generally detrimental effects observed with the aforementioned zero-shot CoT approach from our initial experiments.

Table 13: Comprehensive Performance Comparison on **GEN** Task with direct and CoT Prompting. The best value is highlighted in blue, and runner-up value is highlighted in light blue.

| Type | Model | Embedding | | Keywords | | | N-gram | | | |
|------|-------|-----------|----|----------|--------|------|--------|----------|----------|--------|
| | | SR↑ | SP↑ | Prec.↑ | Recall↑ | F1↑ | BLEU↑ | METEOR↑ | ROUGE-L↑ | Failed↓ |
| | | | | | **One-shot CoT Prompting** | | | | | |
| | o3-mini (OpenAI, 2025) | 0.3064 | 0.2571 | 0.1049 | 0.1990 | 0.1289 | 5.05 | 22.29 | 11.23 | 0.00% |
| | GPT-4o (Achiam et al., 2023) | 0.3544 | 0.2913 | 0.1498 | 0.2849 | 0.1846 | 6.59 | 24.99 | 13.80 | 0.00% |
| Closed | GPT-4-turbo (OpenAI, 2023) | 0.3287 | 0.3571 | 0.1490 | 0.2828 | 0.1837 | 6.81 | 24.47 | 14.99 | 0.00% |
| source | Claude-3-7 (Anthropic, 2025) | 0.4312 | 0.2354 | 0.1556 | 0.2951 | 0.1918 | 6.64 | 25.32 | 13.15 | 0.00% |
| | Gemini-2.0 (Google AI Blog, 2024) | 0.3422 | 0.3156 | 0.1489 | 0.2835 | 0.1834 | 6.68 | 24.88 | 14.22 | 0.00% |
| | Gemini-2.5 (Google AI Blog, 2025) | 0.3773 | 0.2598 | 0.1446 | 0.2746 | 0.1781 | 4.84 | 23.06 | 10.13 | 0.00% |
| | QwQ-32b (Qwen Team, 2025) | 0.2694 | 0.3221 | 0.1433 | 0.2715 | 0.1763 | 5.91 | 22.96 | 12.83 | 0.00% |
| Open | Qwen2.5-72b (Qwen Team, 2024) | 0.3850 | 0.3045 | 0.1573 | 0.2982 | 0.1937 | 7.36 | 25.84 | 14.75 | 0.00% |
| source | Deepseek-V3 (Liu et al., 2024a) | 0.3583 | 0.3031 | 0.1596 | 0.3041 | 0.1971 | 6.57 | 25.59 | 14.83 | 0.00% |
| | Deepseek-R1 (Guo et al., 2025) | 0.3422 | 0.2633 | 0.1320 | 0.2515 | 0.1627 | 4.01 | 19.72 | 8.07 | 0.00% |

Notably, several models demonstrated improvements with one-shot CoT over both direct prompting and zero-shot CoT in specific metrics. For instance, Claude-3-7-sonnet achieved the highest overall *SR* score, reaching **0.4312** with one-shot CoT, a slight improvement over its direct prompting score (**0.4280**) and a significant recovery from its zero-shot CoT performance (**0.3918**). It also saw its *METEOR* score increase to **25.32**, surpassing its direct prompting result (**24.78**). GPT-4-turbo, under one-shot CoT, exhibited the best overall *SP* score (**0.3571**) and the best *ROUGE-L* score (**14.99**), also securing the runner-up position for *BLEU* (**6.81**) in this condition. Deepseek-V3 excelled in keyword-based metrics, achieving the highest *Keyword F1* (**0.1971**) and *Keyword Precision* (**0.1596**) in the one-shot CoT setting. It also performed strongly in N-gram metrics, obtaining the runner-up *METEOR* score (**25.59**) and runner-up *ROUGE-L* (**14.83**). Qwen2.5-72b-instruct demonstrated significant gains, improving its *SR* to **0.3850** (the runner-up score, from 0.3723 in direct) and achieving the overall best *METEOR* score (**25.84**) and the best *BLEU* score (**7.36**) across all models in this one-shot CoT condition (METEOR improved from 23.97 in direct).

However, the improvements were not universal. While *METEOR* scores generally saw a modest uplift with one-shot CoT compared to direct prompting for several models (e.g., GPT-4o to **24.99**, Gemini-2.0-flash to **24.88**), other N-gram metrics like *BLEU* and *ROUGE-L* typically remained lower than their direct prompting counterparts across many models, albeit often showing an improvement over the zero-shot CoT results. Similarly, *Keyword F1* scores, while sometimes better than zero-shot CoT, did not consistently surpass direct prompting levels. For example, Claude-3-7-sonnet's *Keyword F1* of **0.1918** with one-shot CoT was below its direct prompting score of **0.2392**.

These findings suggest that providing even a single, task-relevant example (one-shot CoT) can help models better structure their reasoning process for the **GEN** task, mitigating some of the performance degradation seen with untuned, zero-shot CoT. The exemplar appears to guide the models towards more semantically relevant (*SR*, *METEOR*) and precise (*SP*) step generation for certain architectures. This reinforces the notion that the structure and guidance of the reasoning process are critical for complex generation tasks, and that even minimal exemplars can offer benefits, lending further support to the value of structured CoT exemplars for fine-tuning in the future, as provided by BioProBench.

### G.5 PERFORMANCE COMPARISON ACROSS OUTSTANDING LLMS

The radar charts, as shown in Figure 9 reveal clear distinctions among leading models across comprehension, error detection, ordering and generation tasks. In the **PQA/ERR/ORD** subplot, Gemini-2.5-pro-exp consistently dominates: it achieves the highest PQA accuracy, leads in error-correction precision and overall accuracy, and outperforms peers on both *Exact Match (EM)* and *Kendall's* $\tau$ in the **ORD** task. GPT-4-turbo offers a strong balance between PQA and ERR-*Prec.*, but lags behind Gemini on recall and ordering metrics. Deepseek-R1 closes the gap substantially, particularly on ERR-*Recall* and ORD-$\tau$, demonstrating that an open-source model can rival closed-source alternatives in nuanced judging and local ordering. In contrast, BioMedGPT-LM-7B underperforms across all axes, underscoring its limited procedural understanding despite its biomedical focus.

In the **GEN** subplot, all models exhibit a trade-off between structural fidelity (*Step Recall*) and lexical metrics (*BLEU, METEOR*). Again, Gemini-2.5-pro-exp achieves the best structural recall

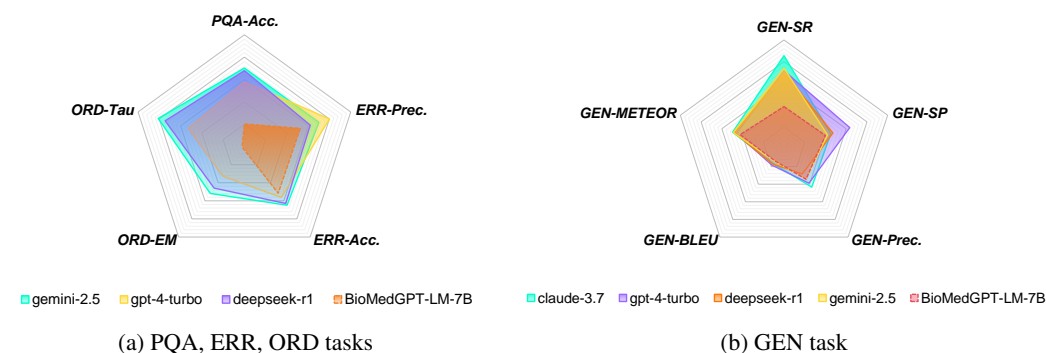

Figure 9: Radar plots comparing performance of representative LLMs on the five core tasks.

and competitive (***Step Precision (SP)***), indicating more complete and less extraneous step genera-
tion. Deepseek-R1 strikes a middle ground with balanced keyword precision and ***METEOR*** scores,
while BioMedGPT-LM-7B trails in every metric, confirming that specialized biomedical pretrain-
ing alone is insufficient for coherent, procedure-aware text generation. Overall, these results high-
light Gemini-2.5's leadership in both reasoning and generation, the surprising competitiveness of
Deepseek-R1, and the pronounced weaknesses of current domain-specific small models.

### G.6 PERFORMANCE OF DOMAIN-SPECIFIC BIO-LLMS

To offer a comprehensive and fair evaluation, this section provides a dedicated analysis of
domain-specific models: BioMedGPT-10B (Luo et al., 2023), BioMistral-7B, and BioMistral-7B-
DARE (Labrak et al., 2024). These models differ significantly from the general-purpose LLMs in
the main text in terms of parameter scale (7-10B) and training corpora (specialized biomedical liter-
ature). Our goal is to understand their performance profile on the procedural tasks in BioProBench,
which present a different set of challenges compared to tasks involving declarative knowledge ex-
traction from scientific articles.

Table 14 and Table 15 summarize their performance. Our analysis indicates that the complex proce-
dural reasoning and generation required by BioProBench are challenging for models of this archi-
tecture and training paradigm.

Table 14: Performance summary for domain-specific Bio-LLMs on comprehension, ordering, and
correction tasks. These models are analyzed separately to fairly account for their distinct scale and
training focus. The up-arrow (↑) denotes that higher is better, while the down-arrow (↓) denotes that
lower is better. A dash (-) indicates the metric was not applicable due to format adherence issues.

| | **PQA** | | | **ORD** | | | **ERR** | | | |
| --- | --- | --- | --- | --- | --- | --- | --- | --- | --- | --- |
| **Model** | *Acc.*↑ | *BS*↓ | *Failed*↓ | *EM*↑ | *τ*↑ | *Failed*↓ | *Acc.*↑ | *Prec.*↑ | *Recall*↑ | *F1*↑ |
| BioMedGPT-10B | 0.202 | 0.535 | 53.4% | 0.020 | 0.020 | 91.2% | 0.515 | 0.526 | 0.273 | 0.359 |
| BioMistral-7B | 0.211 | 0.571 | 33.8% | 0.042 | 0.044 | 91.7% | 0.498 | 0.498 | 0.572 | 0.532 |
| BioMistral-7B-DARE | 0.279 | 0.567 | 37.7% | 0.064 | 0.049 | 64.9% | 0.501 | 0.500 | **0.831** | **0.624** |

On comprehension-heavy tasks like **PQA** and **ORD**, the models struggled to parse the questions and
reconstruct procedural logic, as evidenced by low accuracy and high rates of non-compliance with
output formats. This suggests that understanding the implicit causality and temporal dependencies
in protocols remains a significant hurdle. In the **ERR** task, BioMistral-7B-DARE, which leverages
the DARE model merging technique, achieved a remarkably high ***Recall*** (**83.1%**), indicating a
potentially valuable risk-averse strategy for error detection, despite low precision.

The **GEN** task, which requires synthesizing a complete protocol, proved particularly demanding
(Table 15). The low scores across both our domain-specific metrics (*SR*, *SP*) and standard N-gram
metrics (*BLEU*, *METEOR*) suggest difficulties in generating semantically relevant steps and main-
taining lexical similarity to reference protocols.

Table 15: Performance of domain-specific models on the **GEN** task.

| Model | Embedding | | Keywords | | | N-gram | | |
|---|---|---|---|---|---|---|---|---|
| | *SR↑* | *SP↑* | *Prec.↑* | *Recall↑* | *F1↑* | *BLEU↑* | *METEOR↑* | *ROUGE-L↑* |
| BioMedGPT-10B | 0.1941 | 0.1999 | 0.1721 | 0.2740 | 0.1967 | 6.84 | 21.18 | 15.15 |
| BioMistral-7B | 0.0335 | 0.0445 | 0.1232 | 0.2170 | 0.1486 | 5.81 | 19.21 | 12.53 |
| BioMistral-7B-DARE | 0.1400 | 0.2434 | 0.2163 | 0.1879 | 0.1756 | 4.66 | 15.12 | 15.90 |

In conclusion, our results highlight that the specialized knowledge from biomedical literature does not directly translate to the procedural understanding required for biological protocols. This is not a limitation of the models themselves, but rather an illustration of the distinct nature of the problem space. We believe this presents a valuable opportunity for the community. BioProBench can serve as a crucial resource for fine-tuning and developing the next generation of Bio-LLMs, specifically enhancing their grasp of procedural logic and bringing them closer to practical application in the laboratory.

**Detailed Domain-wise Performance Analysis** A coarse, aggregated evaluation score can mask important variations in model performance across different scientific specializations. To provide a more nuanced understanding of the capabilities of domain-specific LLMs, we conducted a detailed domain-wise performance analysis. Our benchmark was designed with fine-grained domain labels to facilitate this type of granular evaluation.

As an illustrative example, we present results from the **PQA (Protocol Question Answering)** task. Table 16 shows the performance of three prominent domain-specific models across a selection of 14 distinct sub-domains. The accuracy scores clearly demonstrate that model performance is not uniform; different models exhibit unique strengths and weaknesses.

Table 16: Sample sub-domain accuracy (Acc) scores on the PQA task for three domain-specific models. The best-performing model in each sub-domain is highlighted in bold.

| Sub-domain | BioMedGPT-LM-7B | BioMistral-7B | BioMistral-7B-DARE | Key Observation |
|---|---|---|---|---|
| Microbiology & Virology | 0.1750 | 0.3265 | **0.4468** | BioMistral-7B-DARE shows a clear lead. |
| Molecular Biology Techniques | **0.3256** | 0.1897 | 0.2931 | BioMedGPT-LM-7B is surprisingly strong here. |
| Bioinformatics Methods | 0.2727 | **0.3750** | 0.2308 | BioMistral-7B excels in this computational domain. |
| Pharmacology & Drug Dev. | **0.2353** | 0.1667 | 0.1750 | BioMedGPT-LM-7B shows a relative strength. |
| Structural Biology Tech. | 0.1429 | **0.2143** | 0.1176 | BioMistral-7B performs best, though all struggle. |

This granular analysis reveals several key findings:

1. **Specialized Strengths:** Models exhibit distinct performance profiles. For instance, BioMistral-7B-DARE demonstrates superior performance in 'Microbiology & Virology', whereas the base BioMistral-7B model shows a particular aptitude for 'Bioinformatics Methods'. This suggests that training data and fine-tuning methods have a significant impact on sub-domain specialization.

2. **Performance Inversion:** A model's overall aggregated score can be misleading. While BioMistral-7B-DARE may have a strong overall score, BioMedGPT-LM-7B outperforms it in 'Molecular Biology Techniques' and 'Pharmacology & Drug Development', highlighting its specific areas of expertise.

3. **Revealing Common Limitations:** This analysis also pinpoints common weaknesses across models. For example, all models find 'Structural Biology Techniques' challenging, indicating a potential gap in current training datasets or model architectures for this specific area.

This granular evaluation not only strengthens our findings but also provides a much clearer and more valuable picture of the current landscape of domain-specific LLMs, which can help guide future research and development efforts.

G.7 DETAILS OF HUMAN VALIDATION FOR LLM-AS-A-JUDGE

To address concerns regarding the reliability of using an LLM (DeepSeek-V3) as a judge for the Consistency metric in the REA-ERR task, we conducted a rigorous human validation experiment.

The Consistency metric is designed to measure whether the "reasoning chain" generated by a model correctly identifies the specific error described in the Ground Truth. This is a constrained semantic matching task. We randomly sampled 200 instances from the REA-ERR test set where the evaluated models correctly predicted the binary label (True/False).

The PhD-level biological experts were presented with:

1. The Ground Truth Error Description (e.g., "Reagent concentration is too low").

2. The Model's Generated Chain-of-Thought (CoT).

3. The LLM-Judge's decision (Pass/Fail on whether the CoT matched the Ground Truth).

4. The experts were asked to label the LLM-Judge's decision as "Correct" or "Incorrect" based on their domain knowledge.

Finally, the Human-LLM Agreement is 188/200 instances, and the Agreement Rate is 94.21%. The remaining 5.8% of cases were primarily "edge cases" where the model's reasoning was partially correct or ambiguous (e.g., identifying the correct parameter but stating the wrong direction of error). In the vast majority of clear-cut cases, the LLM-judge demonstrated human-level performance in this specific matching task. This result confirms that for the specific purpose of checking reasoning consistency against a known ground truth, the LLM-as-a-judge approach is a reliable proxy for human evaluation.

## G.8 QUALITATIVE ANALYSIS OF REASONING INCONSISTENCY

To address the concern regarding models providing correct answers via incorrect reasoning traces, we analyzed specific instances from the REA-ERR task. A representative example is shown in Table 17, where the model correctly flags the step as erroneous (False) but identifies the wrong scientific cause.

Table 17: Case Study: Hallucinated Error vs. Actual Error.

| | |
|---|---|
| **Target Step (Corrupted):** | *"...centrifuge for 2 min at 10,000× g..."* (Context involves a $10\mu m^3$ sample). |
| **Ground Truth Error:** | The centrifugation speed is too high (10,000× g). The correct protocol specifies 1,000× g. The sample size ($10\mu m^3$) is consistent with the ground truth context. |
| **Model Prediction:** | **[False]** (Correct Label). |
| **Model Reasoning Trace (Excerpt):** | *"Parameter Error: ...'10,000× g': This is a high speed... but could be acceptable for pelleting all cellular material... The most critical error is the sample volume ('10um3')... It is physically impossible to 'dice' a sample of this size...".* |
| **Analysis of Inconsistency:** | The model explicitly condones the actual error (10,000× g) as "acceptable." Instead, it hallucinates a physical impossibility regarding the sample dimensions ($10\mu m^3$), which are actually correct in the specific context of this micro-sample protocol. |
| **Implication:** | Standard accuracy metrics would credit the model for this response. However, our Consistency metric (LLM-as-a-judge) correctly penalizes this instance because the reasoning does not align with the ground truth error (Speed vs. Volume). This case exemplifies why evaluating the reasoning chain is essential for rigorous scientific benchmarking. |

## H EXAMPLES OF LLM RESPONSES

This section provides qualitative examples illustrating the performance of evaluated Large Language Models on select instances from the **BioProBench** benchmark. Complementing the quantitative results presented elsewhere, these examples offer valuable insights into the nature of the task inputs, the expected Ground Truth outputs, and the types of responses generated by different models across various biological protocol understanding and reasoning tasks. For each selected task instance, the input prompt provided to the model is displayed alongside its corresponding ground truth answer and the outputs produced by several representative LLMs from our evaluation suite. Visual cues, such as background coloring, are employed within the accompanying table to highlight the agreement or

disagreement between model responses and the Ground Truth. Representative examples spanning different task types are compiled and presented in Table 18 20 19 21 22 23.

Table 18: Examples illustrating the input, ground truth, and outputs from selected models for PQA task. Model output cells are colored green if they exactly match the Ground Truth and red otherwise.

| Input / Prompt: | **Question:** Place a droplet (100ul) of water-suspended fixated nematodes onto the LbL-coated glass slides and wait ____ min for settling and attaching the animals to the LbL polyelectrolyte film. **Choices:** ["50","20","10","40","30"] **Task Instructions:** Choose the most likely correct answer from the given choices. Output format: [ANSWER_START]choice & confidence[ANSWER_END] |
|---|---|
| **Ground Truth:** | 30 |
| **GPT-4o Output:** | 30 & 80 |
| **Deepseek-v3 Output:** | 20 & 70 |
| **Gemini-2.5-pro-exp Output:** | 30 & 75 |

Table 19: Examples illustrating the input, ground truth, and outputs from selected models for ERR task. Model output cells are colored green if they exactly match the Ground Truth and red otherwise.

| Input / Prompt: | **Question:** Determine whether the following target step in a protocol is True or False: Resuspend the cell pellets in 10mL of culture media (RPMI 1460 containing 10% FBS, 100U/mL of penicillin/strepto-mycin, 2 mM lglutamine, and 500mM beta-mercaptoethanol). **Context:** purpose: Prepare culture media for bone marrow cell suspension., prior_step: Red blood cell lysis and washing., next_step: Addition of GM-CSF for cell culture. **Task Instructions:** Carefully evaluate if the step is logically consistent... Respond with only True or False... Output format: [ANSWER_START]True or False[ANSWER_END] |
|---|---|
| **Ground Truth:** | False |
| **GPT-4o Output:** | False |
| **Deepseek-v3 Output:** | False |
| **Gemini-2.5-pro-exp Output:** | False |

Table 20: Examples illustrating the input, ground truth, and outputs from selected models for ORD task. Model output cells are colored green if they exactly match the Ground Truth and red otherwise.

| Input / Prompt: | Please sort the following steps titled 'Cell Washing and Red Blood Cell Lysis' in the correct order. The steps are: Wash the spleen cells suspension with complete RPMI medium by centrifugation at 350xg for 10 minutes at 4°C. Stop the lysis reaction by adding complete RPMI medium to a final volume of 10ml. Discard the supernatant and resuspend cells in 3ml of ice-cold ACK to lyse red blood cells. Incubate for 5 minutes at room temperature with occasional shaking. - Give me the correct order of the steps as a list of their original indices (start from 0), no other words. - Output your answer wrapped exactly between the tags [ANSWER_START] and [ANSWER_END]. - The format of your response must be: [ANSWER_START]a list of the original indices[ANSWER_END] |
|---|---|
| **Ground Truth:** | [0, 2, 3, 1] |
| **GPT-4o Output:** | [2, 3, 1, 0] |
| **Deepseek-v3 Output:** | [0, 2, 3, 1] |
| **Gemini-2.5-pro-exp Output:** | [0, 2, 3, 1] |

# I  SENSITIVITY ANALYSIS ON DATA CONTAMINATION

A potential concern in evaluating LLMs on open-source data is the risk of data contamination (memorization of training data). To investigate this, we constructed a Perturbed Test Set consisting of 100 randomly selected PQA instances.

For each instance, we systematically modified the questions in the source context. We then re-evaluated the top-performing models (Gemini-2.5-Pro, DeepSeek-R1, o3-mini, DeepSeek-V3) on this perturbed set.

If models were relying on parametric memory (memorization), we would expect them to ignore the modified context and output the original (now incorrect) answers, leading to a significant perfor-

Table 21: Examples illustrating the input, ground truth, and outputs from selected models for GEN task. Model output cells are colored green if they exactly match the Ground Truth and red otherwise.

| Input / Prompt: | **Systemn Role:** You are an expert in CRISPR-based molecular biology protocols, skilled in breaking down complex procedures into atomic steps. 
 **Instruction:** Please describe in a single-level list format (using only 1., 2., 3. numbers, no sub-steps) of the protocol. Only the operating steps should be included, no explanation of the principles or material list is required. The steps should be logically coherent, the language should be concise, and it should follow the structure of a standard experimental protocol. 
 **Format Requirements:** 
 - Each step must be on a separate line. 
 **Question:** To design sgRNAs for targeting genes in a CRISPR competition assay, what are the key steps? |
|---|---|
| Ground Truth: | 1. Identify target sites within the gene of interest. 
 2. Design sgRNAs with 20 base pairs of homology and a NGG PAM sequence. 
 3. Clone sgRNAs into a guide vector backbone using BsmBI digestion and ligation. 
 4. Use online tools like Benchling to optimize sgRNA design. 
 5. Target functional protein domains for efficient gene knockout. |
| GPT-4o Output: | 1. Identify the target genes of interest. 
 2. Select appropriate genomic regions within the target genes for sgRNA design. 
 3. Use sgRNA design tools to generate potential sgRNA sequences. 
 4. Filter the sgRNA sequences based on criteria such as on - target efficiency and off - target potential. 
 5. Choose a set of sgRNAs for the CRISPR competition assay. 
 6. Synthesize the selected sgRNAs. |

mance drop. However, as shown in Table 24, the performance decline was minimal (approximately 0-3%).

The stability of model performance across perturbed contexts strongly suggests that the models are performing in-context reasoning to extract information from the provided text, rather than recalling memorized answers. This validates that our benchmark effectively evaluates reasoning capabilities.

## J DATASHEET FOR BIOPROBENCH

Following the framework proposed by Gebru et al (Gebru et al., 2021), we provide a datasheet to document the motivation, composition, collection process, and maintenance plan of BioProBench.

### J.1 MOTIVATION

For what purpose was the dataset created? The dataset was created to evaluate and improve the procedural reasoning capabilities of Large Language Models (LLMs) in the domain of biological experimental protocols. It aims to address the gap in existing benchmarks which focus primarily on declarative knowledge rather than procedural execution.

Who created the dataset? The dataset was created by the authors of this paper.

### J.2 COMPOSITION

What do the instances that comprise the dataset represent? The dataset consists of two parts:

BioProCorpus: 26,933 full-text biological protocols (raw text and structured JSON).

BioProBench Task Data: Over 550,000 structured task instances (QA pairs, ordering sequences, etc.) derived from the corpus.

Does the dataset contain all possible instances or is it a sample? It is a sample of publicly available protocols from six major repositories (Bio-protocol, Protocol Exchange, etc.), covering 16 biological sub-domains.

Is the information missing from some instances? Some instances derived from restrictive licenses (e.g., JOVE) are excluded from the public release to comply with licensing terms, as detailed in Section 2.

Does the dataset contain data that might be considered confidential or sensitive? No. All data is sourced from scientific protocols that are either open-access or publicly viewable. Personal data of protocol authors has been removed or is publicly cited scientific information.

### J.3 COLLECTION PROCESS

How was the data associated with each instance acquired? Data was collected via web scraping and API access from six specific repositories detailed in the main text.

What mechanisms or procedures were used to collect the data? We used custom Python scripts to crawl and parse the raw HTML/PDF content into a unified JSON format.

Who was involved in the data collection process? The authors and their research assistants.

### J.4 PREPROCESSING/CLEANING/LABELING

Was any preprocessing/cleaning/labeling of the data done? Yes. We used a multi-stage pipeline:

Cleaning: Removal of HTML tags and non-textual artifacts.

Structuring: Using LLMs (LLaMA/DeepSeek) to parse steps into structured JSON.

Task Generation: Programmatically generating task instances (PQA, ORD, etc.).

Quality Filtering: Automated structural checks and manual validation by PhD-level experts.

Is the software used to preprocess/clean/label the instance available? Yes, the code for the construction pipeline is available in our repository.

### J.5 USES

Has the dataset been used for any tasks already? Yes, it is used in this paper to benchmark 10 LLMs and to evaluate the ProAgent.

What (other) tasks could the dataset be used for? It can be used for training scientific agents, improving procedural extraction, and checking bio-safety compliance.

### J.6 DISTRIBUTION

Will the dataset be distributed to third parties? Yes, the openly licensed portion (approx. 380k instances) is available on [GitHub/HuggingFace Link available upon publication].

How will the dataset be distributed? It is distributed as JSON files via the repository.

When will the dataset be distributed? It is available now (anonymized) and will be permanently available upon publication.

License: The public subset is released under CC BY 4.0.

### J.7 MAINTENANCE

Who is supporting/hosting/maintaining the dataset? The corresponding author and their lab.

How can the owner/curator of the dataset be contacted? Contact information will be provided in the final version of the paper.

Is there an erratum? We will maintain a changelog on the repository to document any updates or error corrections.

Will the dataset be updated? Yes, we plan to update the dataset if new protocols become available or if community feedback identifies issues.

Table 22: Examples illustrating the input, ground truth, and outputs from selected models for REA-GEN task. Model output cells are colored green if they exactly match Ground Truth and red otherwise.

| | |
|---|---|
| **Input / Prompt:** | **System Role:** You are an expert in CRISPR-based molecular biology protocols, skilled in breaking down complex procedures into atomic steps. 
 **Instruction:** Please describe in a single-level list format (using only 1., 2., 3. numbers, no sub-steps) of the protocol. Only the operating steps should be included, no explanation of the principles or material list is required. The steps should be logically coherent, the language should be concise, and it should follow the structure of a standard experimental protocol. 
 **Response Structure Requirements:** 
 Your response must be structured strictly for machine processing. It must contain two main parts in order: 
 1. Your Chain of Thought (CoT) process, formatted with specific XML-like tags. 
 2. The final detailed protocol steps, wrapped in `[ANSWER_START] [ANSWER_END]` tags. 
 **CoT Instructions & Format:** 
 Please begin your response by outputting your thinking process. Follow this exact structure and include your analysis within the respective tags: 
 Let's think step by step: 
 `<Objective>`[Output the core objective of this protocol here]`</Objective>`. 
 To achieve this, `<Precondition>`[Output the necessary preconditions, materials, equipment, etc., here]`</Precondition>`. 
 The protocol must proceed as `<Phase>`[Output the logical division into key phases or stages here]`</Phase>`, where critical parameters are `<Parameter>`[Output the critical parameters for each step/phase and the logic behind them here]`</Parameter>`. 
 Finally, `<Structure>`[Acknowledge and state the required output structure for the final steps here]`</Structure>`. 
 After outputting the complete thinking process exactly as structured above, output the final detailed protocol steps. 
 **Final Output Step Format Requirements:** 
 - Each step must be on a separate line. 
 - `[ANSWER_START]` [Output the detailed protocol steps here, ensuring each step is on a new line] `[ANSWER_END]` 
 **Question:** 
 To design sgRNAs for targeting genes in a CRISPR competition assay, what are the key steps? |
| **CoT:** | Let's think step by step: 
 First, ¡Objective¿designing sgRNAs for targeting genes in a CRISPR competition assay¡/Objective¿. 
 To achieve this, ¡Precondition¿the gene sequence of interest is available, and the researcher has knowledge of CRISPR components and tools¡/Precondition¿. 
 The protocol must proceed as ¡Phase¿1) Identifying target sites within the gene, 2) Designing sgRNAs with specific parameters, 3) Cloning sgRNAs into a vector, and 4) Optimizing sgRNA designs¡/Phase¿, where critical parameters are ¡Parameter¿a 20 base pair homology region, presence of an NGG PAM sequence, use of BsmBI digestion for cloning, and targeting functional protein domains for efficient knockout¡/Parameter¿. 
 Finally, ¡Structure¿the protocol is presented as a single-level list with imperative verbs, maintaining logical order and excluding rationale or material lists¡/Structure¿. |
| **Ground Truth:** | 1. Identify target sites within the gene of interest. 
 2. Design sgRNAs with 20 base pairs of homology and a NGG PAM sequence. 
 3. Clone sgRNAs into a guide vector backbone using BsmBI digestion and ligation. 
 4. Use online tools like Benchling to optimize sgRNA design. 
 5. Target functional protein domains for efficient gene knockout. |
| **GPT-4o Output:** | Let's think step by step: 
 ¡Objective¿Design sgRNAs for targeting genes in a CRISPR competition assay¡/Objective¿. 
 To achieve this, ¡Precondition¿Materials required: genomic sequence of target genes, CRISPR design tool (e.g., CHOPCHOP, CRISPRscan), computational access for design, primers, cloning materials, and control sequences (e.g., non-targeting sgRNA)¡/Precondition¿. 
 The protocol must proceed as ¡Phase¿1. sgRNA design and selection, 2. Cloning of sgRNA into a CRISPR vector, 3. Preparation of cells for transfection, 4. Transfection of CRISPR constructs into cells, 5. Competition assay setup and analysis¡/Phase¿, 
 where critical parameters are ¡Parameter¿1. Selection of high-efficiency, on-target sgRNAs, 2. Ensuring proper cloning of sgRNAs, 3. Efficient transfection of CRISPR constructs, 4. Monitoring competition conditions, 5. Validating sgRNA effects through analysis (e.g., flow cytometry, sequencing)¡/Parameter¿. 
 Finally, ¡Structure¿Detailed protocol steps will be listed in a logical sequence with concise, actionable instructions for each phase¡/Structure¿. 
 [ANSWER] 
 1. Obtain the genomic sequences of the target genes to be edited. 
 2. Use a CRISPR design tool to generate potential sgRNA sequences for each target gene. 
 3. Evaluate the sgRNAs based on predicted on-target efficiency and off-target effects. 
 4. Select the top-performing sgRNAs for cloning into a CRISPR vector. 
 5. Synthesize the selected sgRNAs using a commercial service or custom oligos. 
 6. Clone the sgRNAs into the CRISPR expression vector using standard cloning protocols (e.g., ligation or restriction enzyme-based cloning). 
 7. Verify the correct insertion of sgRNAs into the vector via sequencing or restriction digestion. 
 8. Prepare cells (e.g., HEK293T or other relevant cell line) by growing them to the desired confluence. 
 9. Transfect cells with the CRISPR vector carrying the selected sgRNAs using a transfection reagent (e.g., Lipofectamine or electroporation). 
 10. Select successfully transfected cells using appropriate selection markers (e.g., puromycin) or by flow cytometry if using fluorescent markers. 
 11. Set up the CRISPR competition assay by co-transfecting cells with a mix of different sgRNA constructs or using a control sgRNA. 
 12. Allow cells to proliferate under selective conditions to enable competition between edited populations. 
 13. Harvest cells at specified time points and analyze the relative abundance of different populations by flow cytometry, qPCR, or sequencing. 
 14. Analyze the data to assess the competition between gene-edited populations and determine the effect of each sgRNA on gene function. 
 [/ANSWER] |

Table 23: Examples illustrating the input, ground truth, and outputs from selected models for REA-ERR task. Model output cells are colored green if they exactly match Ground Truth and red otherwise.

| Input / Prompt: | **Main Instruction:** Evaluate the validity of the following target step in a protocol. Follow the detailed reasoning process demonstrated in the example below to identify potential errors across Operation, Reagent, and Parameter categories, with meticulous attention to numerical values and their consistency with the provided context and typical practices.
**— Example Start —**
**Example Target Step:**
Mix 860μL of sterile deionized water and 14μL of 5% sodium hypochlorite in a 1.5mL tube.
**Example Context:**
{
"purpose": "Sterilization of seeds to remove surface contaminants using sodium hypochlorite.",
"prior_step": "1.1 Place transgenic Arabidopsis seeds in a 1.5mL tube.",
"next_step": "1.2.2 Vigorously mix the contents of the tube using a vortex mixer."
}
**Example Reasoning Process:**
1. Operation Error: The operations (Mix) and the use of a 1.5mL tube are standard. No obvious operational errors.
2. Reagent Error: The reagents are appropriate. However, the specified volume of 5% sodium hypochlorite is 14μL, mixed with 860μL water. This results in a very dilute solution (0.07%). For sterilization, typical practice suggests a final concentration of around 0.5–1% sodium hypochlorite. Therefore, the reagent volume is significantly too low, which undermines effectiveness and contradicts the stated sterilization purpose.
3. Parameter Error: Although explicit parameters like time and temperature are not mentioned, the concentration of sodium hypochlorite functions as a critical parameter in disinfection efficacy. Here, the final concentration (0.07%) is too low to be effective, making it a parameter error as well.
Based on the significant numerical error in both Reagent volume and the effective concentration (parameter), the step is invalid.
**Example Answer:**
[ANSWER_START] False [ANSWER_END]
**— Example End —**
**Now, evaluate the following target step using the same detailed reasoning process demonstrated in the example above:**
**Evaluate the validity of the target step:**
Maintain hESCs in Nutristem medium on $0.5u/cm^2$ vitronectin N-precoated tissue culture-treated plastic in a humidified incubator ($37°C$, $10\%CO_2$).
**You may use the following context, which includes the purpose of the target step, as well as the preceding and following steps, to inform your decision:**
**Context:**
purpose: Maintain hESCs in optimal conditions for growth and differentiation.
prior_step: "Preparation of vitronectin-coated plates"
next_step: Daily medium exchange
**Analysis Instructions:**
Analyze the step, paying meticulous attention to all numerical values (e.g., times, temperatures, volumes, concentrations, speeds, durations), by reasoning through the following three categories of potential errors. As part of this analysis, explicitly compare numerical values specified in the target step and consider typical laboratory practices.
**Constraints on Evaluation:**
Only evaluate the correctness of the information explicitly present in the target step. Do not make assumptions about missing details. Focus solely on identifying errors in what is actually stated.
**Final Answer Format:**
The format of your final answer must be:
[ANSWER_START] True or False [ANSWER_END] |
| **CoT:** | CO2 concentration is critical for maintaining pH balance in the medium. The concentration of $CO_2$ should be 5%. |
| **Ground Truth:** | False. |
| **o3-mini Output:** | Reasoning:
1. Operation Error:
The instruction "Maintain hESCs in Nutristem medium on . . . tissue culture-treated plastic" is generally appropriate as an operation for culturing human embryonic stem cells. The use of pre-coated plastic (vitronectin-coated) and a humidified incubator at $37°C$ is standard practice. There is no error in the overall procedure described.2. Reagent Error:
The medium (Nutristem) and the coating (vitronectin) are appropriate reagents for hESC culture. The target step states the coating density as "$0.5u/cm^2$." This appears to be intended to represent a concentration value (likely 0.5 $μg/cm^2$, a typical value) but is rendered in an unusual format. However, when considering only the explicit numerical value and comparing to typical practices, 0.5 $μg/cm^2$ is standard. Since we are evaluating the correctness of the explicit information and assuming that the shorthand "$0.5u/cm^2$" is meant to denote this, the reagent information is acceptable.
3. Parameter Error:
The incubator condition is stated as "$37°C, 10\%CO_2$." While $37°C$ is the standard temperature, the typical $CO_2$ concentration for culturing hESCs is around 5% in most conventional cell culture protocols. A setting of 10% $CO_2$ is unusually high and could adversely affect the pH and cell viability. This numerical discrepancy in the $CO_2$ parameter represents an error relative to standard practices.Based on the significant error in the $CO_2$ parameter, the target step is not valid.
[ANSWER_START]False[ANSWER_END] |

Table 24: Sensitivity Analysis on Data Contamination

| Model | Original Acc. | Perturbed Acc. | $\triangle$ Acc. | $\triangle$ Brier Score |
|---|---|---|---|---|
| Gemini-2.5-Pro | 0.71 | 0.69 | -0.02 | 0.058 |
| DeepSeek-R1 | 0.67 | 0.64 | -0.03 | 0.066 |
| o3-mini | 0.66 | 0.63 | -0.03 | 0.022 |
| DeepSeek-V3 | 0.65 | 0.63 | -0.02 | 0.01 |

