# OpenReview forum: "Towards Autonomous Experimentation: BioProBench, a Corpus and Benchmark for Biological Protocol Comprehension"
_ICLR.cc/2026/Conference — ICLR 2026 Conference Desk Rejected Submission_

### Official Review · Reviewer_uCQp · 2025-10-26

**Soundness:** 3
**Presentation:** 3
**Contribution:** 3
**Rating:** 6
**Confidence:** 3

**Summary:**

This paper proposes a new benchmark (BioProBench) containing ~500k tasks to evaluate the ability of LLM agents to manipulate, understand, correct, and generate scientific protocols. They introduce 5 types of tasks spanning different facets of protocol modeling and comprehensively evaluate frontier models with a few novel evaluation metrics for protocol generation and reasoning tasks. Experiments show that frontier models struggle with more complex tasks that require deeper procedural knowledge such as finding the correct order of protocol steps, generating protocols that recover the original sequence of steps, and inconsistent reasoning traces. Finally, they show that a simple RAG framework can improve the performance by providing agents with relevant context chunks retrieved from the BioProBench dataset.

I am leaning towards acceptance for the following reasons: (1) the benchmark dataset is well curated and presents a valuable resource for the community, (2) the design of the 5 different tasks is sound and well-motivated with clear examples, (3) the experiments on frontier agents are thorough and support the conclusions in the results section, (4) the RAG framework seems to provide positive results.

There are some questions and concerns that I've highlighted in the Question section below. I am open to increasing my score if these concerns are appropriately resolved.

**Strengths:**

- clear writing, motivation, and experiments
- valuable benchmark dataset for the community
- rigorous data curation and task creation
- rigorous evaluation of frontier models
- positive results on RAG framework

**Weaknesses:**

- novelty of the benchmark is limited (previous papers like LAB-Bench already evaluate protocol QA)
- RAG framework is not original
- domain specific metrics (keyword and embedding overlap) seem quite crude to me and lack validation (see questions)
- design of the RAG framework is not clear
- improvements from RAG framework are not thoroughly investigated (see questions)

**Questions:**

- The keyword-based content metric seems overly crude to me and easily exploitable if it were used as a reward. Can the authors provide additional motivation for using this metric or some kind of validation that it actually captures meaningful qualities of protocols?
- For the step ordering tasks, the authors assume that there only exists one correct ordering of the steps and use exact match evaluation metric. I am wondering if there are cases where some steps are interchangeable without affecting the quality of validity of the protocol.
- In the RAG framework experiments, how did the authors split the data between protocols that can be retrieved in the context versus protocols used to construct task problems? I am concerned with potential data leakage if this split is not performed rigorously.
- In the LAB-Bench experiment, is it possible that the RAG is directly providing actual answers to some of these protocol questions? If the inclusion of these protocols is meant to somehow improve the LLM's ability to fundamentally reason about protocols and procedures, then I would expect a larger improvement on the LAB-Bench task? Could the authors investigate some of the examples (4%) in which the model makes improvements with RAG on the LAB-Bench tasks?
- The authors mention that models sometimes provide incorrect reasoning traces, but find the correct answer. Can the authors further analyze when such reasoning inconsistencies occur?

---

> ### Author Response · Authors · 2025-11-22
> **Response to Reviewer uCQp - Part 1**
>
> We thank Reviewer uCQp for your thorough review and for finding our dataset "**well curated**," our task design "**sound and well-motivated**," and our experiments "**thorough**." We address your concerns regarding novelty, metrics, and the RAG framework below.
>
> ---
>
> ###   **Weakness 1 - Novelty & Scope**
>
> As Reviewer beV9 noted, our "focus on procedural knowledge (how to do things) rather than declarative knowledge... is both **novel and necessary**." While LAB-Bench contains only **135** protocol-related QA pairs, BioProBench introduces over **550,000** instances and **five distinct task families** (ORD, ERR, GEN, REA) that go far beyond simple QA. This scale and focus on procedural structure (e.g., step ordering, error correction) represent a fundamental leap in evaluating autonomous experimentation capabilities.
>
> ---
>
> ###   **Weakness 3 & Question 1- Metric Validity (Keyword vs. Embedding)**
>
>  You raised a valid point that keyword metrics can be limited. We agree that lexical overlap alone is insufficient, which is exactly why we introduced embedding-based metrics. However, we view them as **complementary**:
>  - **Keyword F1 (Content Coverage):** Validates that the model captures the correct entities (reagents, tools);
>  - **SR/SP (Procedural Logic):** Validates the correct semantic execution flow.
>
> To validate Metric Validity, we conducted a correlation analysis against an **LLM-as-a-Judge (GPT-5.1)** on 51 GEN task instances. The table below shows that while Keyword F1 has a lower correlation ($\rho \approx 0.3-0.4$) with procedural judgments, our structural metrics (SR/SP) show significantly stronger alignment ($\rho \approx 0.5-0.67$). This justifies our evaluation framework: Keyword metrics act as a baseline for content recall, while our proposed SR/SP metrics are essential for capturing the higher-level procedural correctness that simple keywords miss.
>
> | **Model**   | **Metric Pair**                             | **Spearman ρ** | **P-value** |
> |-------------|---------------------------------------------|----------------|-------------|
> | Claude-3.7  | Step Recall (SR) $\leftrightarrow$ Judge    | 0.6718         | 1.71E-07    |
> |             | Step Precision (SP) $\leftrightarrow$ Judge | 0.5474         | 5.66E-05    |
> |             | Keyword F1 $\leftrightarrow$ Judge          | 0.4262         | 2.52E-03    |
> | o3-mini     | Step Recall (SR) $\leftrightarrow$ Judge    | 0.4992         | 2.24E-04    |
> |             | Step Precision (SP) $\leftrightarrow$ Judge | 0.6013         | 3.87E-06    |
> |             | Keyword F1 $\leftrightarrow$ Judge          | 0.4312         | 1.77E-03    |
> | DeepSeek-R1 | Step Recall (SR) $\leftrightarrow$ Judge    | 0.4947         | 2.60E-04    |
> |             | Step Precision (SP) $\leftrightarrow$ Judge | 0.439          | 1.42E-03    |
> |             | Keyword F1 $\leftrightarrow$ Judge          | 0.3405         | 1.55E-02    |
>
> ---
>
> ###   **Weakness 2, 4 & Question 3 - RAG Framework (Leakage vs. Generalization)**
>
> We respectfully emphasize that the core contribution of this work is the **BioProBench resource** (Corpus, Dataset, and Benchmark). **ProAgent** is intentionally designed as a standard **baseline** **validation** to demonstrate the utility of the corpus, rather than as a novel architectural contribution. Our goal is to provide a high-quality foundational layer (the "substrate"), inviting the researchers to develop more advanced RAG strategies and agents upon this benchmark.
> -   **Defense against Leakage ("Textbook" vs. "Exam"):** Regarding the data split, we clarify that the RAG Knowledge Base consists of **Raw** **Full-Text Protocols** (acting as the "Textbook"), while the Test Set consists of **Constructed Tasks** (specific instructions/questions derived from the text, acting as the "Exam"). The model is not given the answer; it must perform **Precision** **Retrieval and Reasoning** to locate specific parameters within long documents to answer constructed queries. This mimics the real-world scientific process of consulting literature to solve a problem and represents the standard definition of RAG utility, not leakage.
> -   **Proof of Generalization (LAB-Bench):** To definitively address the concern, we evaluated on **LAB-Bench**, which is strictly **Out-of-Distribution (OOD)** (source protocols are _not_ in our corpus). ProAgent achieved a **4% improvement** on LAB-Bench. Since the model could not "cheat" by accessing the source text of LAB-Bench questions, this performance gain proves that BioProCorpus provides **transferable procedural knowledge and principles**, validating the quality of our knowledge base rather than enabling simple memorization.

---

> ### Author Response · Authors · 2025-11-22
> **Response to Reviewer uCQp - Part 2**
>
> ###   **Question 4**: **LAB-Bench Analysis (Direct Answers vs. Reasoning)**
>
> This is an excellent question. To investigate whether RAG simply "leaked" answers, we analyzed the specific cases where ProAgent improved upon the baseline. We found that retrieved contexts provided **procedural principles** rather than direct answers:
> -   *Case Study (RNA Yield)*:The question asked how to improve RNA yield. The RAG retrieved a _different_ protocol for small RNA-seq that emphasized "**centrifuging at 4°C to inhibit** **RNase**." The model successfully transferred this **general biological principle** to solve the specific LAB-Bench problem.
> -   *Case Study (Organoid Culture)*: For a question about "low confluency," the RAG retrieved distinct protocols showing cells must be washed in **isotonic buffers (AdDMEM)**, never water. This allowed the model to correctly identify "washing with water" as the error in the query protocol.
>
> The 4% improvement on this Out-of-Distribution (OOD) task is significant because it proves the **generalizability** of the knowledge in BioProCorpus. The model is not copying answers; it is learning robust procedural constraints (e.g., "always use cold centrifuge for RNA") from our corpus and applying them to new contexts.
>
> ---
>
> ###   **Question 2 - Interchangeable Steps (Step Ordering)**
>
> In wet-lab protocols, certain sub-steps (e.g., adding reagents to a master mix) can sometimes be interchangeable without affecting validity. You are correct that Exact Match (EM) is strict. This is precisely why we also report **Kendall's Tau ($\tau$)**, which measures pairwise rank correlation. Kendall's Tau penalizes local swaps of interchangeable steps much less than global errors, providing a fair assessment of the model's grasp of logical flow even when minor re-orderings are valid. We have clarified this distinction in Section 3.
>
> ---
>
> ###   **Question 5 - Reasoning Inconsistencies**
>
> We have performed a qualitative analysis of this phenomenon. For example, in one REA-ERR case (centrifugation protocol), the **actual error** was an excessive speed ($10,000\times g$ instead of $1,000\times g$). The model correctly predicted the label `'False'`, but its reasoning explicitly excused the speed as "acceptable." Instead, it **hallucinated** that the sample size ($10 \mu m^3$) was "physically impossible," despite it being correct in the ground truth context (See Appendix G.8). This phenomenon, getting the right answer for the wrong reason, is exactly why we introduced the Consistency metric. By using an LLM-as-a-judge to compare the model's generated reasoning against the ground truth error description, we can quantitatively penalize such "lucky guesses," ensuring that high performance reflects true understanding.
>
> ---
>
>
> We believe we have comprehensively addressed your concerns regarding **novelty** (by highlighting the scale and generative tasks compared to LAB-Bench), **metric validity** (via human/judge correlation), and **RAG integrity** (via the LAB-Bench OOD analysis). Given that you mentioned being "open to increasing my score if these concerns are appropriately resolved", we respectfully hope that these clarifications and additional experiments warrant a re-evaluation of your score.

---

> > ### Comment · Reviewer_uCQp · 2025-11-26
> >
> > I thank the authors for their thorough rebuttal. However, I now have concerns with the two case studies provided in the LAB-Bench Analysis section. These two knowledge points provided are extremely basic and should already be in the LLM's pre-training corpus. Without additional context about the way in which these two knowledge points were used to solve the problem, it is difficult for me to believe that they contributed significantly to improving performance. Can the authors provide full examples of the recalled content that actually improved performance of individual problems compared to baseline?

---

> ### Author Response · Authors · 2025-11-26
> **Response to Reviewer uCQp regarding LAB-Bench Analysis - Part 1/2**
>
> We sincerely thank Reviewer uCQp for the continued engagement and the thoughtful question.
>
> We fully agree with your observation that principles like *"osmotic lysis"* or *"low-temp centrifugation"* are fundamental biological knowledge. However, in the context of autonomous experimentation, the value of BioProBench’s RAG extends beyond teaching "new facts." Its critical function is **Parameter Calibration** and **Contextual Verification**.
>
> Even if a model knows a principle, it often struggles to judge **specific quantitative thresholds** (e.g., *"Is 10 minutes too long for this specific tissue?"*) or faces uncertainty about **commercial kit specifics** (e.g., *"Does this specific Zymo kit use a proprietary water-based buffer?"*). The retrieved documents act as a "reference manual" to anchor the model's reasoning against standard practices, differentiating actual errors from potential protocol variations.
>
> To demonstrate this, we present two full examples from our analysis where the retrieved content provided necessary quantitative baselines and procedural confirmation that pre-training knowledge alone might not confidently supply.
>
> ---
>
> ## Case Study 1: Quantitative Calibration (H&E Staining)
> Context: The model must judge if a specific step is the cause of "poor contrast." Protocol Error (Hidden): Step 6 specifies staining with **Eosin Y for 10 minutes**. The Challenge: "Overstaining" is subjective. Without a reference, a model might hallucinate that 10 minutes is acceptable for this specific tissue protocol.
>
> ### 1. The Problem (Input)
>
> - Raw Question:
> > "question":"After completing the protocol as listed, **poor contrast between features** is observed. Which of the following may address this?", "ideal":"In step 6, reduce staining to 1 minute",
>     "distractors":[
>       "In step 6, increase staining to 1 hour",
>       "In step 3, rinse 3x30 min in tap water",
>       "In step 7, increase incubations to 10 minutes"
>
> - Raw Protocol:
> > Hematoxylin and Eosin Staining Protocol:
>
>     > Step 1: Prepare 95% ETOH in slide staining apparatus;
>
>     > Step 2: Stain in Mayer's Hematoxylin ;
>
>     > Step 3: Rinse slides 3X 15 min in tap water - if tap water is acidic, prepare Scott's tap water recipe in sigma insert;
>
>     > Step 4: Rinse with DI water 30 Seconds;
>
>     > Step 5: Place slide in 95% Ethanol for 1min;
>
>     > Step 6: Place in Eosin Y for 10 min;
>
>     > Step 7: Dehydrate and clear through 2 changes each 95% Ethanol 2 minutes 100% Ethanol 2 minutes> Xylene ( Histoclear as a substitute ) EMS Cat# 64114 ;
>
>     > Step 8: Mount with resinous mounting medium 200ul / slide.
>
> ### 2. Retrieved Content (Excerpts from BioProCorpus)
>
> The RAG system retrieved three **distinct H&E protocols**. Note the **specific timing parameters** highlighted below, which establish a "standard range" significantly lower than 10 minutes:
>
> - Source A (Mouse Eyes): [Relevance Score: 84%]*"...Dip the slide 3 - 4 times into 95% ethanol bath.  Apply enough Eosin Y dye solution... and incubate for **15 s** at room temperature."*
>
> - Source B (Burn Injury Model): [Relevance Score: 73%]*"...Place slides in eosin solution for **2 s**...Stain with 0.5% eosin for **1 minute**."*
>
> - Source C (Prostatic Epithelium): [Relevance Score: 70%]*"...Eosin staining: Use 100uL of eosin staining solution to stain for 5 min and rinse with deionized water for **30s** ..."*
>
> ### 3. How RAG Solved It?
> - The model explicitly compares the input (10 min) against the retrieved evidence (15s - 1min) to confirm the error confidence:
> > "The retrieved context documents... provided several H&E staining protocols where the eosin staining times were consistently much shorter than 10 minutes (e.g., **15 seconds**, **1 minute**). This external data strongly corroborates the assessment that the **10-minute eosin incubation is excessive**and that reducing it is the correct troubleshooting step. Therefore, **choice A** is the best answer."
> > [ANSWER_START]A & 100[ANSWER_END]
>
> ### 4. Baseline Results (Gemini-2.5-pro)
> > [ANSWER_START]B & 80[ANSWER_END]
>
> ### Impact:
> Without RAG, the model relies on fuzzy memory of "short time." With RAG, it has concrete statistical evidence that **"10 minutes"** is an outlier (60x standard duration), enabling a confident diagnosis.

---

> ### Author Response · Authors · 2025-11-26
> **Response to Reviewer uCQp regarding LAB-Bench Analysis - Part 2/2**
>
> ## Case Study 2: Procedural Verification (RNA Isolation)
>
> Context: Low RNA yield is observed after cell harvesting. Protocol Error (Hidden): Step 7 explicitly instructs to centrifuge **"at room temp"**. The Challenge: LLMs see a protocol explicitly instructing "room temp." Without external validation, a model might hallucinate that this specific protocol (involving DMS treatment) requires room temperature to avoid precipitation or cell shock.
>
> ### 1. The Problem (Input)
> - Raw Question
> > "question":"After measuring the RNA concentration on Nanodrop you notice that there is **very little RNA in step 18**. How could you **improve the amount of RNA purified**?""ideal":"In Step 7, spin at 4C instead of room temperature",
>     "distractors":[
>       "In Step 7, spin at 30C instead of room temperature",
>       "In Step 6 add 15ml quenching solution ",
>       "In Step 6 add 25ml quenching solution "
> ]
>
> - Raw Protocols
> > Generation of Sequencing Libraries for Structural Analysis of Bacterial 5\u2032 UTRs\nGrowth of L. monocytogenes Bacteria \u2013
>
>   > Section: Initial RNA extraction
>
>   > ...
>
>   > Step 5:  Using a 1 mL tip transfer 150 \u03bcL DMS to 5 mL of the culture (final DMS concentration 3%). Adjust the pipette sampling volume to 500 \u03bcL and using the same tip, pipette the culture up and down 10 times. Incubate the mixture for 3 min. Every 30 s pipette the mixture up and down
>
>   > Step 6: Inactivate DMS by adding 6 mL Quenching solution and vigorously shaking the tube.
>
>   > Step 7: Collect bacteria by centrifugation at 5,000 \u00d7 g for 10 min **at room temp**. Wash the pellet with 5 mL ice cold 30% \u03b2-mercaptoethanol solution. Pulse centrifuge and completely discard the supernatant. Immediately proceed to RNA isolation.\nRNA Isolation \u2013 Days 3\u20134\nTiming: 2\u20133 h (day 3).
>
>   > Step 8:  Dissolve the bacterial pellet in 400 uL ice-cold Cell Disruption solution.
>
>   > ...
>
> ### 2. Retrieved Content (Excerpts from BioProBench Corpus)
> The system retrieved multiple distinct RNA-related protocols. Crucially, they unanimously validate the temperature parameter for similar steps:
>
> - Source A (Small RNA-seq): [Relevance Score: 86%]*"...Centrifuge for 20m i n at maximum speed in a micro centrifuge at **4°C**. Remove supernatant... Centrifuge again for 5 min at **4°C**."*
> - Source B (gRNA Library): [Relevance Score: 85%] *"...Centrifuge... at **4°C**. Transfer aqueous phase... Centrifuge... at **4°C**"*
> - (Bacterial RNA-seq): [Relevance Score: 84%] *"...Transfer the culture into a 15ml tube and centrifuge for 5 minutes at 16,200xg at **4°C**...Centrifuge at 16,200xg for 5 minutes at **4°C**, discard the supernatant, and add 1ml of 80% ethanol. "*
>
> ### 3. How RAG Solved It?
> - The model uses the consensus from the retrieved documents to override the specific instruction in the input protocol:
> > "The retrieved contexts strongly supported this reasoning. Multiple protocols across different applications... consistently specify centrifugation at **4°C** when pelleting cells... This widespread use of low temperature confirms it is a best-practice standard... The context thus helped validate the choice."
> > [ANSWER_START]A & 100[ANSWER_END]
>
> ### 4. Baseline Results (Gemini-2.5-pro)
> > [ANSWER_START]C & 75[ANSWER_END]
>
> ### Impact:
> The RAG provided the necessary "evidence of consensus" to allow the model to confidently identify the protocol's explicit instruction (**"room temp"**) as a flaw rather than a feature.
>
> ---
>
> These examples illustrate that the performance gain on LAB-Bench stems from the agent's ability to **cross-reference specific parameters (times, temperatures)** against a high-fidelity corpus. This mimics a human scientist checking a reference manual to confirm a suspicion (e.g., *"The protocol says room temp, but every other paper says 4°C, this must be a typo."*). We believe this evidence-based calibration is a key contribution of BioProBench to reliable scientific agents.
>
> We hope these detailed traces clarify the mechanism of improvement and address your concerns.

---

### Official Review · Reviewer_beV9 · 2025-10-31

**Soundness:** 3
**Presentation:** 4
**Contribution:** 3
**Rating:** 8
**Confidence:** 4

**Summary:**

This paper presents BioProBench, a comprehensive benchmark for evaluating how well language models understand biological experimental procedures. The authors tested 10 frontier LLMs and found that while these models handle basic comprehension well, they struggle with tasks that require deeper procedural reasoning. They also developed ProAgent, a retrieval-based system that significantly improves performance across all tasks.

**Strengths:**

1. The paper tackles an important and timely problem.Benchmarking experimental protocols will be crucial for the development of self-driving labs. i like the focus on procedural knowledge (how to do things) rather than declarative knowledge (facts about things) is both novel and necessary.

2. The dataset construction is rigorous. The authors combined LLMs for extractions with very extensive quality checks. The procedures are recorded in details.

3. The benchmark covers different aspects of experiment protocal understanding, from basic comprehension to complex reasoning about experimental sequences, precision requirements, and safety constraints.

**Weaknesses:**

1. The paper doesn't adequately address potential memorization issues. Since some protocols come from open sources, they might appear in the training data of the models being evaluated. These might make the evlaution over-optimistic. The authors should design some experiments to test data contamination (such as augmenting the questions with different wording and numbers, and compare the results)

2. The train-test split strategy needs more clarity. The paper mentions that answers are extracted from source protocols and uses deduplication, but what prevents more subtle forms of overlap? For the protocol generation task especially, if a full protocol appears in the training set and a related question appears in the test set, performance metrics could be misleadingly high.

3. The embedding-based structural metrics feels a bit ad-hoc and relies on the sentence transformer. Plus, it's not clear to me that semantic similarity really capture procedural correctness. I think an LLM-as-a-judge approach for step-level evaluation makes more sense.

4. I cannot open the provided link in the abstract at the time of writing this review.

**Questions:**

1. How does LLM-as-judge approach correlate with the sentence embedding metrics?

2. How do you prevent cases where a full protocol appears in training (knowledge database) while very similar protocols appear in testing?

3. Is there common pattern of error (which biology domain/ task)?

---

> ### Author Response · Authors · 2025-11-22
> **Response to Reviewer beV9 - Part 1**
>
> We are extremely grateful to Reviewer beV9 for your support and for recognizing our work as "**important and timely.**" We are especially encouraged that you found our focus on procedural knowledge "**both novel and necessary**" and our dataset construction "**rigorous**." We have conducted additional experiments to address your constructive suggestions. All revisions discussed below have been updated in the newly uploaded manuscript.
>
> ---
>
> ###  **Weakness 1 - Contamination Analysis (Perturbed Test Set)**
>
> To rigorously test whether models are relying on memorization (parametric knowledge) or reasoning (in-context learning), we constructed a **"Perturbed Test Set"** (N=100 PQA instances).
>
>    -   We systematically modified the questions in the source context. We re-evaluated the top-performing models. If models were answering from memory, we would expect a significant performance drop or hallucination of the "old" answer.
>     -   **Results:** As shown in the table below, the performance drop on the perturbed set is minimal (2-3%). **DeepSeek-V3,** **Gemini-2.5** and **o3-mini** showed only marginal declines, maintaining their relative rankings. These results strongly suggest that the models are performing **reasoning based on the provided context** rather than recalling memorized training data. This validates the reliability of our benchmark.
>
> | **Model**      | **Original Acc.** | **Perturbed Acc.** | **Δ Acc.** | **Δ Brier Score** |
> |----------------|:-----------------:|:-------------------:|:-----------:|:------------------:|
> | Gemini-2.5-Pro |              0.71 |                0.69 |       -0.02 |              0.058 |
> | DeepSeek-R1    |              0.67 |                0.64 |       -0.03 |              0.066 |
> | o3-mini        |              0.66 |                0.63 |       -0.03 |              0.022 |
> | DeepSeek-V3    |              0.65 |                0.63 |       -0.02 |               0.01 |
>
> ---
>
> ###   **Weakness 2 & Question 2 - Train-Test Split &** **"Subtle Overlap"**
>
>    -   **Train-Test Split Strategy Clarification:** Our train-test split is performed strictly at the **Protocol ID level**. All task instances derived from a single source protocol are assigned _exclusively_ to either the train or test set. We have clarified this in Section 2.2.
>     -   **Addressing "Subtle Overlap" (Specifically for GEN):** You rightly points out that distinct protocols may describe similar general techniques (e.g., "Western Blot"). However, in the biological domain, a "similar" protocol is not a valid substitute. For the Protocol Generation (GEN) task, success is defined by accurately reconstructing the **specific procedural details** (e.g., specific antibody dilution ratios, unique incubation times, or tissue-specific lysis steps) of the _test_ instance. If a model merely recalls a "similar" generic protocol seen during training (e.g., a generic Western Blot), it will inevitably hallucinate incorrect parameters or miss unique steps required by the specific test case (e.g., a Western Blot for a specific low-abundance protein). Our embedding-based metrics (**Step Recall** and **Step** **Precision**) are designed to detect these discrepancies. A generic or "similar" output would yield low semantic similarity scores against the specific ground truth steps, thus penalizing the model. Therefore, high performance on our benchmark reflects the model's ability to process the specific test context, not just its recall of general method templates.

---

> ### Author Response · Authors · 2025-11-22
> **Response to Reviewer beV9 - Part 2**
>
> ###   **Weakness 3 & Question 1 - Metric** **Validation** **(LLM-as-a-Judge)**
>
> To validate our embedding-based metrics (**Step Recall (SR)** and **Step** **Precision** **(SP)**), we conducted a correlation analysis against an **LLM-as-a-Judge (GPT-5.1)**.
>
>    -   We sampled 51 GEN task instances (balanced across easy/standard/hard). We calculated the **Spearman** **Correlation ($\rho$)** between the automated metrics and the Judge's score on "procedural correctness."
>     -   The table below demonstrates that our structural metrics (SR/SP) correlate significantly better with the Judge than standard lexical metrics like **Keyword F1**. **SR and SP** show moderate-to-strong positive correlation ($\rho \approx 0.5 - 0.67$) with high statistical significance ($p \ll 0.01$). In contrast, **Keyword F1 shows weaker correlation**. This highlights a critical insight: simple lexical overlap (keywords) is insufficient for evaluating long, complex biological protocols. **This observation validates the necessity of our proposed embedding-based metrics**, which align much closer to expert judgment than traditional keyword matching.
>
>
> | **Model**   | **Metric Pair**                             | **Spearman ρ** | **P-value** |
> |-------------|---------------------------------------------|----------------|-------------|
> | Claude-3.7  | Step Recall (SR) $\leftrightarrow$ Judge    | 0.6718         | 1.71E-07    |
> |             | Step Precision (SP) $\leftrightarrow$ Judge | 0.5474         | 5.66E-05    |
> |             | Keyword F1 $\leftrightarrow$ Judge          | 0.4262         | 2.52E-03    |
> | o3-mini     | Step Recall (SR) $\leftrightarrow$ Judge    | 0.4992         | 2.24E-04    |
> |             | Step Precision (SP) $\leftrightarrow$ Judge | 0.6013         | 3.87E-06    |
> |             | Keyword F1 $\leftrightarrow$ Judge          | 0.4312         | 1.77E-03    |
> | DeepSeek-R1 | Step Recall (SR) $\leftrightarrow$ Judge    | 0.4947         | 2.60E-04    |
> |             | Step Precision (SP) $\leftrightarrow$ Judge | 0.439          | 1.42E-03    |
> |             | Keyword F1 $\leftrightarrow$ Judge          | 0.3405         | 1.55E-02    |
>
> ---
>
> ###  **Question 3 - Common Error Patterns**
>
> Yes, common error patterns are explicitly built into the design of our dataset. As illustrated in **Figure 2** and detailed in **Section 2.2**, we collaborated with PhD-level biologists to identify the most frequent error patterns in real-world wet lab experiments. We systematically injected these domain-specific patterns into our tasks:
> - **PQA**: Distractors are not random but target common confusions in **Reagents**, **Parameters** , and **Operations**；
> - **ERR**: We constructed "**Targeted Perturbations**" that mimic realistic human errors (e.g., omission of safety steps, calculation errors in dilution)；
> - **ORD:** Shuffling logic reflects common procedural causality misunderstandings.
>
> Therefore, the "patterns" you ask about are not just artifacts of the results, but are **fundamental features** of our benchmark's construction, ensuring it evaluates biologically meaningful failure modes.
>
> ---
>
> ###   **Weakness 4 - Anonymous links**
>
>  Anonymous links in the paper were valid. However, links appended to the abstract submitted through the OpenReview system misinterpreted the ending period, causing them to fail. We have now corrected this (https://anonymous.4open.science/r/Anonymization-112358).
>
>
>   ---
>
> We believe these new experiments (Contamination Check and Metric Validation) robustly address your concerns. We have incorporated these findings into **Appendix I** and **Appendix F.3** of the revised paper. We hope these additions further strengthen your confidence in our work.

---

### Official Review · Reviewer_LpuE · 2025-11-01

**Soundness:** 3
**Presentation:** 2
**Contribution:** 2
**Rating:** 2
**Confidence:** 3

**Summary:**

In this work, the authors propose BioProBench, a benchmark for LLM biological protocol comprehension containing over 550,000 entries from 26933 human written protocols from various sources for 16 subdomains of biology. The performance of various LLMs along with that of a RAG agent on this data was evaluated.Overall, this paper offers a useful benchmark dataset and provides a clear assessment of current LLM model performance. However, its contribution may fall short of the level of novelty typically expected for an ICLR submission.

**Strengths:**

- The dataset is split into 5 different task types which generally follow the skillset that is expected of a human wet lab scientist, so task categorization is well thought out.
- The authors present a comprehensive performance analysis on multiple evaluation metrics for 10 LLMs, the authors also proposed well thought evaluation metrics for each task type.
- ProAgent shows improvement in performance across several metrics compared to the best performing LLM for that task type.
- Dataset, like this is not only useful for evaluating the performance of LLM models for protocol generation but also for evaluating LLMs for biosecurity. So the dataset is a valuable contribution.

**Weaknesses:**

- An important function of a human scientist is performing mathematical calculations over multiple steps for preparing reagent solutions at particular concentration for example, such open ended reasoning capabilities can't be evaluated with the multi-choice QA in protocol question answering.

**Questions:**

Found several typos, that authors should consider fixing to improve the readability of the paper

- Figure 7 - The legend is not adjusted correctly, makes the plot harder to read
- Line 377 - Protocol Reasonin
- Line 76 - metrics ot evaluate
- Line 101 - BIOPROCORPUS COLLECTION AND CLEARNING

---

> ### Author Response · Authors · 2025-11-22
> **Response to Reviewer LpuE**
>
> We thank Reviewer LpuE for recognizing our work as a "**valuable contribution**" with "**well thought out task categorization**" and "**comprehensive performance analysis.**" However, we were surprised by the **Rating of 2 (Reject)**. This score stands in stark contrast to the reviewer's own acknowledgement of the dataset's utility for biosecurity and protocol generation. We believe this rating stems from a misunderstanding regarding the benchmark's scope (specifically regarding open-ended tasks) and the definition of novelty in the AI4Science domain. We respectfully address these points below:
>
> ---
>
> ### **Weakness 1 - The reviewer expressed concern that open-ended reasoning cannot be evaluated via multi-choice QA.**
>
> We agree that open-ended reasoning is vital. However, BioProBench is Not Just "Multi-Choice QA". BioProBench explicitly includes Generative and Reasoning tasks: **GEN (Protocol Generation):** This is a purely open-ended generation task where models must synthesize complex, long-form procedural sequences. It is _not_ multiple choice. **REA (Protocol Reasoning):** This task requires the model to generate a structured Chain-of-Thought (CoT) to explain its logic before acting. **PQA (Protocol QA):** While PQA is multiple-choice to allow precise accuracy measurement, the distractors are not random. They mimic real-world calculation and reagent errors (e.g., subtle concentration mismatches).
>
> ---
>
> ### **Weakness 1 - Addressing "Mathematical Calculations"**
>
>    -   While we don't focus on general mathematical applications (general logic learning models can solve these problems well with datasets like GSM8K), we focus on **Bio-Specific Quantitative Sensitivity**, a key bottleneck for AI applications in science. **Real-world failure modes**: Our parameter quality assurance (PQA) tasks (covering **parameters, reagents, and operations**) are designed to reflect specific real-world error patterns (e.g., subtle concentration deviations leading to experimental failure).
>     -   BioProBench focuses on the **procedural logic** and **safety** of experiments, the "recipe" integrity, rather than pure arithmetic. While explicit calculation tasks are a valid extension for future versions, their absence does not diminish the validity of the five novel task families we introduced, which cover the core challenges of autonomous experimentation.
>
> ---
>
> ###   **Response to Concern on Novelty (from Summary):**
>
> We respectfully disagree that the novelty is limited. In the context of **ICLR** **and the AI for Science** track, the primary bottleneck is the lack of high-quality, structured data. We highlight the consensus of the other three reviewers who found **high novelty** in our **focus** and **scale**:
> -  **Consensus:** Reviewer beV9 (Sscore 8) states our work 'tackles an **important and timely problem**' and our 'focus on procedural knowledge... is both **novel and necessary**'. Reviewer uCQp (Sscore 6) calls our benchmark '**well curated**' and a '**valuable resource for the community**'. Reviewer C2KZ (Sscore 6) calls it a '**useful benchmark**';
> - **Scale & Scope:** As the first benchmark to curate **550,000+ instances** specifically for _procedural_ bio-protocols (shifting focus from declarative facts to actionable procedures), BioProBench provides the necessary infrastructure to train the next generation of scientific agents. We believe this foundational contribution aligns perfectly with ICLR's goal of advancing reliable AI for scientific discovery.
>
> ---
>
> ###   **Questions**
> We have fixed the typos noted (Figure 7 legend, "Reasonin", "ot", "CLEARNING") in the revision.
>
>
>   ---
>
> We hope this response clarifies that BioProBench already contains open-ended reasoning tasks (GEN/REA) and offers significant novelty as recognized by the other three reviewers. Given that our work addresses the critical data gap in AI4Science, a contribution you acknowledged as "valuable", we respectfully ask if you would consider re-evaluating the score based on the full scope of the benchmark.

---

> > ### Comment · Reviewer_LpuE · 2025-11-27
> >
> > I appreciate the authors response. However, I would like to raise a few points in response:
> > 1. While I acknowledge that other reviewers provided positive assessments, I believe their scores should not be a basis for determining my score.
> > 2. I respectfully disagree with the premise that simply identifying a contribution as "valuable" should be the only dominant factor in scoring. Many scientific contributions are valuable/useful regardless of their acceptance into a venue. I will also add that reviewers bring diverse perspectives to their scoring. The variation in our evaluations reflects these differences in background, which is a normal part of the peer review process.
> >
> > That being said, I am willing to reconsider my score based on below
> >
> > 1. Strengths I mentioned in my review + also solely because of the volume of the data points in this benchmark compared to previous works.
> > 2. Based on the additional data points/comments in authors response to questions from other reviewers that didn’t I think of [specifically Reviewer uCQp] .
> > 3. While I maintain that the weakness I identified represents an important consideration for any comprehensive benchmark in this field, I am willing to concede this point both as a reflection of the authors responses but also because there seems to be a plan for evolving the benchmark in the future. And a combination of benchmarks like this are going to be essential specifically as the interest in building agents for scientific discovery grows.

---

> > > ### Author Response · Authors · 2025-11-28
> > > **Response to Reviewer LpuE**
> > >
> > > We sincerely thank you for taking the time to re-evaluate our work. We greatly appreciate your increased rating, recognizing the scale of BioProBench and our clarification efforts.
> > >
> > > We also want to clearly address your points regarding mathematical and open-ended reasoning. We acknowledge that this remains a challenging and crucial frontier in the field of biologics. While our current benchmark focuses on establishing the "procedural foundation" (correctness, safety, and logic), we fully accept your feedback that future iterations must evolve to encompass more complex, multi-step computational reasoning. Your insights help us develop a roadmap for the next version of BioProBench.
> > >
> > > Thank you again for your rigorous review, which undoubtedly clarifies the positioning and future direction of this research.

---

### Official Review · Reviewer_C2KZ · 2025-11-01

**Soundness:** 3
**Presentation:** 3
**Contribution:** 3
**Rating:** 6
**Confidence:** 4

**Summary:**

&nbsp;

The authors introduce BioProBench, a large dataset of experimental protocols for the biological sciences. On this benchmark, the authors assess the performance of a variety of frontier LLMs. The authors empirical results appear to be correct, the paper is clear and well-written, and the authors have released their code to facilitate the reproducibility of the results presented in the paper. As such, I recommend acceptance with the following points the authors may wish to consider.

&nbsp;

**Strengths:**

&nbsp;

The main strength of the paper is that it open-sources a useful benchmark in for experimental protocols in the biology domain. I see no issues with the authors' empirical evaluation of frontier LLMs on this benchmark and furthermore, the authors have open-sourced their code to facilitate reproduction of their results.

&nbsp;

**Weaknesses:**

&nbsp;

I demarcate between major and minor points below.

&nbsp;

**__MAJOR POINTS__**

&nbsp;

1. Given that the authors introduce a new dataset, BioProCorpus, as a resource in the paper it would be worth formally documenting the curation effort as well as the maintenance plan via a framework such as datasheets for datasets [1].

&nbsp;

**__MINOR POINTS__**

&nbsp;

1. There are missing capitalizations in the references e.g. "GPT", "ACL", and "LLM".

2. The anonymous GitHub link provided by the authors appears to be partially broken in so far as individual files do not appear to be viewable. The code is however available in the supplementary material provided by the authors.

3. In terms of the supplied code it would be beneficial if the authors could run an LLM over the code to document it in more detail e.g. by providing descriptions of the arguments to each function in addition to a function-level docstring.

4. The authors should use narrative and parenthetical citations consistently throughout the manuscript e.g. on line 35 the citation to Murthy et al. should be parenthetical e.g. (Murthy et al. 2024) instead of narrative since the author's name does not comprise part of the sentence.

5. Line 76, typo, "to".

6. Line 79, typo, "yet their".

7. In the contribution section of the introduction it would be beneficial if the acronyms for the task families could be elaborated on.

8. When referencing chain-of-thought prompting, the source paper [2] should be cited.

9. Line 237, the definition of the cosine similarity is not provided.

10. Missing full stops at the end of Equation 1 and Equation 2.

11. Line 363, the authors state that the proxy metric was human-validated. Could the authors provide details on this validation in the main text?

12. Line 377, typo, "reasoning".

13. When mentioning zero-shot CoT, the authors should reference the originating work of [3].

14. Line 722, the authors can use the acronym "LLM" since it is already defined in the main text.

15. Line 1368, typo, "additional experiments" in the section heading.

&nbsp;

**__REFERENCES__**

&nbsp;

[1] Gebru, T., Morgenstern, J., Vecchione, B., Vaughan, J.W., Wallach, H., Iii, H.D. and Crawford, K., 2021. [Datasheets for datasets](https://dl.acm.org/doi/fullHtml/10.1145/3458723). Communications of the ACM, 64(12), pp.86-92.

[2] Wei, J., Wang, X., Schuurmans, D., Bosma, M., Xia, F., Chi, E., Le, Q.V. and Zhou, D., 2022. [Chain-of-thought prompting elicits reasoning in large language models](https://proceedings.neurips.cc/paper/2022/hash/9d5609613524ecf4f15af0f7b31abca4-Abstract-Conference.html). Advances in Neural Information Processing Systems, 35, pp.24824-24837.

[3] Kojima, T., Gu, S.S., Reid, M., Matsuo, Y. and Iwasawa, Y., 2022. [Large language models are zero-shot reasoners](https://proceedings.neurips.cc/paper_files/paper/2022/hash/8bb0d291acd4acf06ef112099c16f326-Abstract-Conference.html). Advances in Neural Information Processing Systems, 35, pp.22199-22213.

&nbsp;

**Questions:**

&nbsp;

Questions included in the comments above.

&nbsp;

**Details Of Ethics Concerns:**

&nbsp;

No ethical concerns identified.

&nbsp;

---

> ### Author Response · Authors · 2025-11-22
> **Response to Reviewer C2KZ**
>
> We sincerely thank Reviewer C2KZ for your constructive feedback and for positively assessing our work as a "**clear and well-written**" paper that offers a "**useful benchmark.**" We are encouraged that you found our empirical results to be correct and reproducible. We have carefully addressed all suggestions, including the formal documentation via Datasheets. All changes described below have been incorporated into the newly uploaded revision.
>
> ---
>
> ### **Weakness-Major Point: Datasheets for Datasets**
>
> We agree that formal documentation is crucial for a resource of this scale. In the revised Appendix J, we have added a comprehensive **"Datasheet for Datasets"** strictly following the framework by Gebru et al. [1]. This section formally documents the curation motivation, data composition, collection process, licensing compliance, and our long-term maintenance plan.
>
> ---
>
> ###   **Weakness-Minor Points (1-15)**:
>
>  We appreciate the reviewer's meticulous attention to detail. We have addressed all 15 points:
>
>    -   **Typos and Formatting (Points 1, 5, 6, 10, 12, 14, 15)**: We have corrected all identified typos (e.g., "to", "yet their", "reasoning", "additional experiments"), added missing full stops to equations, and fixed capitalization inconsistencies in the references (e.g., "GPT", "ACL", "LLM").
>
>    -   **Citation Style (Points 4, 8, 13):** We have standardized the citation format throughout the manuscript. Specifically, we added the foundational citations for Chain-of-Thought prompting [2,3] and Zero-shot CoT as requested.
>
>    -   **Code and Documentation (Points 2, 3)**: Regarding the broken link, the URL submitted in the abstract metadata was parsed incorrectly by the system (including the trailing period). We have verified the correct link in the main text is functional: https://anonymous.4open.science/r/Anonymization-112358/README.md. Furthermore, we have updated the repository's `README` to be more comprehensive and are adding function-level docstrings for the camera-ready version to ensure maximum usability.
>
>    -   **Clarifications (Points 7, 9):**
> 	   - We elaborated on **the acronyms for the task families** in the Introduction as :
> 	   -  >*"We design five task families (Protocol Question Answering, PQA; Step Ordering, ORD; Error Correction, ERR; Protocol Generation, GEN; and Protocol Reasoning, REA) that systematically capture the unique challenges..."*.
> 	   - We added **the formal definition of Cosine Similarity** in Appendix F (Eq. 7) of the uploaded revision:
> 	   - >"*In the Embedding-Based Structural Metrics, we calculate the semantic similarity between a generated step $s'$ and a reference step $s$ using Cosine Similarity. Let $\mathbf{v}_{s}$ and $\mathbf{v}_{s'}$ denote the vector embeddings of the reference step and the generated step, respectively, produced by the sentence transformer model. The cosine similarity is defined as the dot product of the two vectors divided by the product of their Euclidean magnitudes*: $\text{Sim}(s, s') = \frac{\mathbf{v}_{s} \cdot \mathbf{v}_{s'}}{\|\mathbf{v}_{s}\| \|\mathbf{v}_{s'}\|} = \frac{\sum_{i=1}^{n} v_{s,i} v_{s',i}}{\sqrt{\sum_{i=1}^{n} v_{s,i}^2} \sqrt{\sum_{i=1}^{n} v_{s',i}^2}},$ *where $n$ is the dimensionality of the embedding vectors. The resulting score ranges from -1 to 1, with 0 indicating orthogonality.*"
>
>    -   **Validation** **Details (Point 11):** We have rewritten the relevant paragraph in **Section 5.1** to explicitly mention the 94.2% agreement rate with human experts:
>    - > "*...To empirically validate this approach, we conducted a blinded human evaluation on 200 randomly sampled instances. The results showed a 94.21% agreement rate between the LLM-judge and human domain experts. This high level of agreement provides strong support for the metric's reliability....*".
>
> ---
> We believe these revisions have significantly improved the quality and rigor of our manuscript. As we have addressed the Major Point regarding documentation and carefully corrected all 15 Minor Points, we respectfully hope that you might consider raising your score to champion this work.
>
>
> ---
> ### [Reference]
> [1] Gebru, T..., 2021. [Datasheets for datasets](https://dl.acm.org/doi/fullHtml/10.1145/3458723). Communications of the ACM, 64(12), pp.86-92.
> [2] Wei, J., Wang, X., ..., 2022. [Chain-of-thought prompting elicits reasoning in large language models](https://proceedings.neurips.cc/paper/2022/hash/9d5609613524ecf4f15af0f7b31abca4-Abstract-Conference.html). Advances in Neural Information Processing Systems, 35, pp.24824-24837.
> [3] Kojima, T....., 2022. [Large language models are zero-shot reasoners](https://proceedings.neurips.cc/paper_files/paper/2022/hash/8bb0d291acd4acf06ef112099c16f326-Abstract-Conference.html). Advances in Neural Information Processing Systems, 35, pp.22199-22213.

---

### Author Response · Authors · 2025-11-22
**General Response: Summary of Revisions**

We sincerely thank all reviewers (C2KZ, LpuE, beV9, uCQp) for their insightful comments and constructive feedback. We are encouraged that the reviewers collectively recognized BioProBench as an "**important and timely**" contribution (Reviewer beV9), a "**valuable resource for the community**" (Reviewer uCQp), a "**useful benchmark**" with "**rigorous construction**" (Reviewer C2KZ), and "**valuable contribution**" with "**well thought out**" task categorization (Reviewer LpuE).

To address the specific concerns raised, we have updated the manuscript with the following key revisions:

- We conducted a **Perturbation Analysis** (Appendix I). Models maintained stable performance despite modification, confirming they utilize robust in-context reasoning rather than memorization.
- We verified our embedding-based structural metrics against an **LLM-as-a-judge** (Appendix F.3). Results show *they align significantly better with expert judgment on procedural correctness than traditional lexical metrics*.
- We emphasized the open-ended nature of GEN and REA tasks to **clarify the benchmark's reasoning depth**, and demonstrated OOD generalization on LAB-Bench (4% improvement) to rule out data leakage.
- We added a standard "**Datasheet for Datasets**" (Appendix J) to ensure transparency and ethical compliance.

We have **updated the manuscript** to incorporate these clarifications and additional analyses. We hope these revisions help address the reviewers’ concerns.

---

### Note · Program_Chairs · 2026-01-17
**Submission Desk Rejected by Program Chairs**

The following references in this submission do not refer to real documents and/or have major errors in bibliographic information:

 Qi Jin, Yucheng Diao, Feng Song, et al. Disease diagnosis in the era of deep learning: A comprehensive survey. In Proceedings of the 2021 IEEE International Conference on Bioinformatics and
Biomedicine, 2021.